
# The infrared fixed points of 3d $\mathcal{N} = 4$ $USp(2N)$ SQCD theories

**Benjamin Assel[1][⋆] and Stefano Cremonesi[2][†]**

**1** Theory Department, CERN, CH-1211, Geneva 23, Switzerland
**2** Department of Mathematical Sciences, Durham University, Durham DH1 3LE, UK

⋆ benjamin.assel@gmail.com     † stefano.cremonesi@durham.ac.uk

## Abstract

We derive the algebraic description of the Coulomb branch of 3d $\mathcal{N} = 4$ $USp(2N)$ SQCD theories with $N_f$ fundamental hypermultiplets and determine their low energy physics in any vacuum from the local geometry of the moduli space, identifying the interacting SCFTs which arise at singularities and possible extra free sectors. The SCFT with the largest moduli space arises at the most singular locus on the Coulomb branch. For $N_f > 2N$ (good theories) it sits at the origin of the conical variety as expected. For $N_f = 2N$ we find two separate most singular points, from which the two isomorphic components of the Higgs branch of the UV theory emanate. The SCFTs sitting at any of these two vacua have only odd dimensional Coulomb branch generators, which transform under an accidental $SU(2)$ global symmetry. We provide a direct derivation of their moduli spaces of vacua, and propose a Lagrangian mirror theory for these fixed points. For $2 \leq N_f < 2N$ the most singular locus has one or two extended components, for $N_f$ odd or even, and the low energy theory involves an interacting SCFT of one of the above types, plus free twisted hypermultiplets. For $N_f = 0, 1$ the Coulomb branch is smooth. We complete our analysis by studying the low energy theory at the symmetric vacuum of theories with $N < N_f \leq 2N$, which exhibits a local Seiberg-like duality.

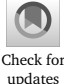
# 1   Introduction and summary of the results

The gauge coupling in three-dimensional Yang-Mills gauge theories has a positive mass dimension, implying that Yang-Mills gauge theories are asymptotically free and, if the number of matter fields is not too large, strongly coupled at low energies. In $\mathcal{N} = 4$ supersymmetric gauge theories the space of vacua is parametrized by continuous moduli. Most of the time it has several components, or branches, comprising the famous Coulomb and Higgs branches. At a generic point on the Coulomb branch the gauge symmetry is broken to a maximal torus and the low-energy theory can be described by free twisted hypermultiplets, while at a generic point on the Higgs branch the gauge symmetry is completely broken in most theories and the low-energy theory can be described by free hypermultiplets.[1] At special locations in the moduli space where new massless degrees of freedom appear, the infrared theory involves an interacting superconformal sector.

It is sometimes believed that at the 'origin' of the moduli space of vacua the theory flows to some SCFT, where 'origin' refers to a point in the moduli space where no nontrivial operator acquires a vev. However this naive picture does not hold for all theories. For instance we may consider the $\mathcal{N} = 4$ pure $SU(2)$ gauge theory. It has only a Coulomb branch, which is isomorphic to the Atiyah-Hitchin manifold [1] and which is smooth. At any point in this moduli space the low-energy theory is that of a free twisted hypermultiplet and there is no interacting SCFT. This happens because part of the classical Coulomb branch is deformed by quantum effects and the so-called origin, where the gauge symmetry is classically enhanced, is lifted.

In the work of Gaiotto and Witten [2] a classification of gauge theories was proposed in terms of the UV R-charges of BPS monopole operators. Monopole operators in the 3d Euclidean theory are local disorder operators whose insertion is defined in the path integral formulation as follows. First we must choose an embedding $\rho : U(1) \hookrightarrow G$, where $G$ is the gauge group. For a given choice of $\mathcal{N} = 2$ subalgebra, this selects an abelian $\mathcal{N} = 2$ vector multiplet $(A_\mu, \lambda, \sigma)$ where $\sigma$ is a real scalar. We then impose a BPS Dirac monopole singularity in the path integral

---

[1]Sometimes there is no genuine Higgs branch, but instead only a mixed branch with a maximal dimensional Higgs factor. Then the low energy theory at a generic point retains some abelian gauge symmetry and has both free twisted and untwisted hypermultiplets.

in the vicinity of the insertion point $x_0 \in \mathbb{R}^3$,

$$dA = \star d\left(\frac{1}{2|x-x_0|}\right), \quad \sigma = \frac{1}{2|x-x_0|}. \tag{1.1}$$

In addition, the monopole insertion can be dressed with an $\mathfrak{h}_\rho$ invariant polynomial $P(\varphi_\rho)(x_0)$ of the $\mathfrak{h}_\rho$ valued adjoint complex scalar $\varphi_\rho$, where $\mathfrak{h}_\rho \subset \mathfrak{g}$ is the subalgebra preserved by $\rho$ (*i.e.* the algebra of the commutant of $\rho(U(1))$ in $G$). This defines a half-BPS monopole operator $V_{\rho,P}$ labelled by the embedding $\rho$ and the dressing polynomial $P$. When the embedding is trivial, $\rho(g) = id$, the operators are simply the gauge invariant polynomials of the adjoint complex scalar, generated by the Casimir invariants like $\text{Tr}(\varphi^n)$. Often one recasts the embedding $\rho$ as a vector of magnetic charges $\vec{m} \in \Lambda_{\text{cochar}}/\mathcal{W}$, where $\Lambda_{\text{cochar}}$ is the cocharacter lattice and $\mathcal{W}$ the Weyl group of the gauge algebra $\mathfrak{g}$, and the monopole operators are labeled $V_{\vec{m},P}$.

BPS monopole operators are chiral operators. Their UV R-charge under the relevant $\mathcal{N}=2$ subalgebra was proposed in [2] based on the earlier work [3],[2] and it was observed that when the number of massless hypermultiplets in the theory is below a certain bound, some monopole operators have R-charges smaller than 1/2, violating naively the 3d unitarity bound in the low energy SCFT. The interpretation in that case is that the R-charge in the infrared theory is not the same as that of the UV theory. Gaiotto and Witten proposed a distinction between *good* theories where the UV R-charges of all monopoles are above the free field value 1/2, *bad* theories where at least one monopole has R-charge below 1/2, and *ugly* theories where some monopoles have R-charge equal to 1/2 and all the others above 1/2. For ugly theories it is then expected that monopole operators saturating the bound become free at low energies.

In good theories, all chiral operators have R-charges above 1/2 and it is expected that the Coulomb branch (as well as the Higgs branch) is an algebraic cone, with an origin (the tip of the cone) corresponding to the vacuum where no operator has a vev and where the theory flows to an interacting SCFT without free fields. This picture has been confirmed by various recent studies of the moduli space [6–14] and agrees with the predictions of mirror symmetry [15–19].

In ugly theories, there are chiral monopole operators of R-charge 1/2 which are expected to become free at low energies [2]. It is expected that the moduli space has flat directions corresponding to free twisted hypermultiplets and that the transverse space to these free flat directions be the moduli space of a good theory. At the origin in this transverse space (and at any location on the flat directions parametrized by the free fields) the infrared theory should be described by an interacting SCFT plus free twisted hypermultiplets.

For bad theories the infrared limit is less clear. In general one might expect SCFTs with free twisted hypermultiplets, however we will see that even this naive expectation turns to be wrong in certain theories at special locations on the Coulomb branch.

In [14] we studied the moduli space of $\mathcal{N}=4$ $U(N)$ SQCD theories with $N_f$ flavours, using the algebraic description of the Coulomb branch proposed in [12].[3] The possible fixed points of good ($N_f \geq 2N$), bad ($N_f \leq 2N-2$) and ugly ($N_f = 2N-1$) theories were studied and it was found that they all fall into a class of fixed points $T_{U(N),N_f}$ with $N_f \geq 2N$, a subclass of the $T_\rho(SU(N_f))$ theories of [2], that one can reach at the origin of the Coulomb branch of good theories. For bad and ugly theories the low energy theory at any point on the Coulomb branch has always decoupled free hypermultiplets, as was naively expected.

In this paper we continue our analysis and study in particular the Coulomb branch of $\mathcal{N}=4$ SQCD theories with gauge group $USp(2N)$ and $N_f$ flavours of hypermultiplets. In the

---

[2]The formula was then proven in [4,5].

[3]The algebraic description of the Coulomb branch of 3d $\mathcal{N}=4$ theories and its quantization were further studied in [20]. The results were recently confirmed by exact supersymmetric localization computations for abelian theories in [21], extending techniques developed in [22] to study the Higgs branch.

classification of Gaiotto and Witten these theories fall only in good and bad classes (no ugly theories), with good theories having $N_f > 2N$ and bad theories having $N_f \leq 2N$. We follow the same approach as in [14], which consists in studying the singularities, or rather singular subvarieties, of the Coulomb branch (CB). The CB singularities arise when matter fields and W-bosons become massless and are the location of infrared interacting fixed points. They are also the locations where Higgs branch factors intersect the Coulomb branch. In general one finds a nested structure of singular subvarieties of increasing quaternionic codimension. By studying the local algebraic geometry near a point in a given singular locus, we can understand the infrared theory in terms of an interacting SCFT and free twisted hypermultiplets. As a result we obtain a classification of the infrared fixed points that one can reach at various locations on the Coulomb branch of $USp(2N)$ SQCD theories.

Our main results are summarized schematically in Figure 1, where we emphasize the low-energy theory at a point in the most singular subvariety $\mathcal{C}^*$ of the Coulomb branch. For good theories, *i.e.* $N_f > 2N$, we find that the most singular locus is a single point, the origin of the Coulomb branch, and that the infrared theory at this point is a certain SCFT that we denote $T_{USp(2N),N_f}$, and which corresponds to the theory $T_{(2(N_f-N)-1,2N+1)}(SO(2N_f))$ in the notation of [2] (see [9] for details). For bad theories with $N_f = 2N$, we find the interesting result that the most singular locus consists of two points, related by a $\mathbb{Z}_2$ global symmetry acting on the Coulomb branch, where a monopole operator takes non-zero vev. The infrared theory at any of these two points is an interacting SCFT that we call $T_{USp(2N),2N}$, and which would be labelled $T_{(2N,2N)}(SO(4N))$ in the notation of [2]. Interestingly there is no decoupled twisted hypermultiplets, despite the fact that the theories are bad. (This is related to the fact that each of the two most singular points is the root of a Higgs branch where the gauge group is completely broken.) In the $N = 1$ case the singularity is an $A_1$ singularity and the infrared SCFTs are $T_{U(1),2} \cong T[SU(2)]$ theories, as already observed in [1]. For $N > 1$, the $T_{USp(2N),2N}$ theories are genuinely new SCFTs. For bad theories with $N_f = 2m+1$ (odd number of flavours), we find that the most singular CB locus has a single extended component and the low-energy theory at a given point is an interacting $T_{USp(2m),2m+1}$ SCFT together with $N-m$ free twisted hypermultiplets. For bad theories with $N_f = 2m < 2N$ (even number of flavours), the most singular locus has two disjoint extended components and the low-energy theory at any point is an interacting $T_{USp(2m),2m}$ SCFT plus $N-m$ free twisted hypermultiplets. When $N_f = 0, 1$ ($m = 0$), the Coulomb branch is smooth and the low-energy theory at any point consists of $N$ free twisted hypermultiplets. Note that all the interacting SCFTs which we find on the Coulomb branch of $USp(2N)$ SQCD with $N_f$ flavours are in the class $T_{USp(2r),N_f}$ with $2r \leq N_f$.

The most striking outcome of this analysis is the existence of the fixed points $T_{USp(2N),2N}$ which do not arise at the origin of the Coulomb branch of good theories, but rather at special locations on the Coulomb branch of the $USp(2N)$ theories with $2N$ flavours, where the monopole operator of vanishing UV R-charge acquire vev. (This is similar in spirit to Argyres-Douglas fixed points [23], which are found at special locations on the Coulomb branch of 4d $\mathcal{N} = 2$ theories.) We find that the fixed point theory has an accidental $SU(2)_J$ symmetry acting on its Coulomb branch. The infrared R-symmetry $SU(2)_{IR}$ is different from the ultraviolet $SU(2)_{UV}$ R-symmetry acting on the full Coulomb branch. More precisely, $SU(2)_{UV}$ can be identified as the diagonal subgroup of $SU(2)_{IR}$ and $SU(2)_J$.

We also revisit the Higgs branch of the $USp(2N)$ SQCD theories. For the $USp(2N)$ theory with $N_f = 2N$, the Higgs branch has two isomorphic components, which classically intersect. We find that in the quantum theory, these two components split, each one getting attached to one of the special singularities associated to the $T_{USp(2N),2N}$ fixed points.[4] This means that the Higgs branch of the $T_{USp(2N),2N}$ SCFT is simply one of the Higgs components of the $USp(2N)$

---

[4]This is similar to the splitting between the baryonic and non-baryonic branches in 4d $\mathcal{N} = 2$ $SU(N)$ SQCD theories, described in [24].

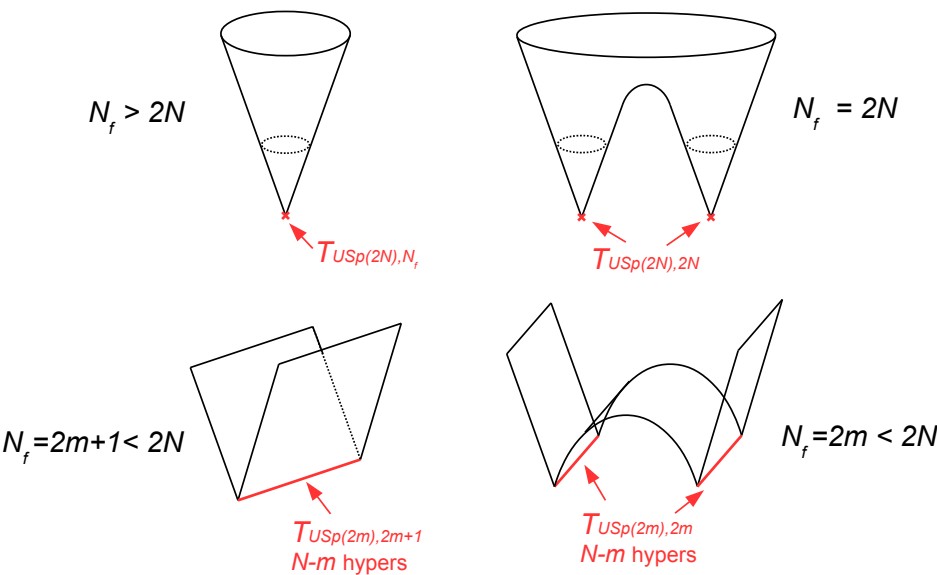

Figure 1: Schematic pictures of the Coulomb branches and effective SCFTs at any point in the most singular subvariety $\mathcal{C}^*$ (in red) for different ranges of $N$ and $N_f$.

gauge theory. Similar observations apply to bad theories with even number of flavours.

It is interesting to note that one could have argued for the existence of the $T_{USp(2N),2N}$ fixed points from mirror symmetry along the lines of [2] section 5.2.2 and section 7. The main idea is that the naive mirror dual of some special good quiver theories, that one deduces from brane constructions, seem to correspond to bad theories. In that case one might expect that the SCFT occurs somewhere on the moduli space of the bad theory. We confirm this scenario by identifying the $T_{USp(2N),2N}$ SCFT as the mirror dual of the balanced D-shaped quiver gauge theory shown in Figure 4, based on the observations of [25]. We identify the moduli space of this SCFT directly from the UV bad gauge theory description. We then provide a very non-trivial check of the mirror symmetry proposal by matching the Coulomb branch Hilbert series of the $T_{USp(2N),2N}$ theory, obtained from the algebraic description of its Coulomb branch, with the Higgs branch Hilbert series of the D-shaped quiver. It is worth noting that the analysis of [2] provides a different (good) mirror dual quiver theory given by $T^{(2N,2N)}(SO(4N))$, in the form of a balanced A-shaped quiver with ortho-symplectic gauge nodes. We will not test this second duality.

Although our analysis in this paper is devoted to the study of the 3d theories, it is interesting to notice that the remarkable physics of the $USp(2N)$ bad theories with even number of flavours is already present in their 4d $\mathcal{N}=2$ parent theories. The 4d $\mathcal{N}=2$ $USp(2N)$ theories with $2n$ flavours ($1 \le n \le N$) were studied in [26] using their Seiberg-Witten curves and differentials.[5] The analysis presented there focused on one maximal singular point on the Coulomb branch, but it is not hard to see that there are actually two such singular points with the same infrared physics.[6] The 4d infrared SCFT at these points is described by a $D_n$ trinion theory (the reduction of the 6d (2,0) $D_n$ type theory on a certain three-punctured sphere) coupled to two hypermultiplets by gauging an $SU(2)$ flavour symmetry, and a free sector of $N-n$ $U(1)$ vector multiplets. This is analogous to what we find in this paper and it is natural to conjecture that

---

[5]We thank S. Giacomelli for discussions on this point.

[6]In the language of [26] the two singular points are obtained at the Coulomb branch parameters $u_i = 0$, for $i \ne N-n+1$, $u_{N-n+1} = \pm 2\Lambda^{2N-2n+2}$, and vanishing complex masses $c_{2i} = \tilde{c}_{2n} = 0$. The SW curve at these points is $y^2 = x^{2n-1}(x^{N-n+1} \mp 4\Lambda^{2N-2n+2})$.

these 4d $\mathcal{N} = 2$ SCFTs reduce on a circle at low energies to the $T_{USp(2n),2n}$ SCFTs with $N - n$ free twisted hypermultiplets, which is the infrared theory at the most singular locus of the 3d $\mathcal{N} = 4$ $USp(2N)$ theory with $2n$ flavours.

Finally we address the question of potential Seiberg-like dualities in 3d $\mathcal{N} = 4$ $USp(2N)$ SQCD theories, by showing that the bad theories with $N + 1 \leq N_f \leq 2N$ admit a *symmetric vacuum*, where no operator takes vev, and where the low energy theory has an interacting $T_{USp(2(N_f - N - 1)),N_f}$ SCFT and $2N - N_f + 1$ free twisted hypermultiplets. We confirm this observation by matching the sphere partition functions of the the bad theory, with the product of the sphere partition function of the good $USp(2(N_f - N - 1))$ theory with $N_f$ flavours and the sphere partition functions of the free hypermultiplets. This infrared relation arises at the symmetric vacuum, but the infrared equivalence does not extend to the full moduli space of the bad $USp(2N)$ theory and therefore we do not refer to it as a duality.

The paper is organised as follows. In Section 2 we review and improve the algorithm of [12] for deriving the algebraic description of the Coulomb branch of 3d $\mathcal{N} = 4$ gauge theories, and we apply it to the $USp(2N)$ SQCD theories. We point out some subtleties related to the elimination of spurious branches. In Section 3 we study the structure of the singular subvariety of the Coulomb branch and analyse the low-energy limit at each of the singular loci. In particular we identify the $T_{USp(2N),N_f}$ SCFTs with $N_f > 2N$ at the CB origin of good theories and the $T_{USp(2N),2N}$ SCFTs at the two special singular points of the bad $USp(2N)$ theories with $2N$ flavours. In Section 4 we review the known analysis of the Higgs branch and explain how it fits with our analysis of the Coulomb branch. In Section 5 we propose a mirror dual realization of the $T_{USp(2N),2N}$ fixed points and perform a test by matching Hilbert series. In Section 6 we study the low-energy physics at the symmetric vacuum, which mimics a Seiberg duality. Our analysis is supported by an identity between exact sphere partition functions. In the appendices we collect various computations and some additional results. Appendix A presents a derivation of the Coulomb branch of the $SU(2) \simeq USp(2)$ SQCD theories as a hyperkähler quotient of the Coulomb branch of the $U(2)$ SQCD theories. This gives an alternative confirmation of the prescription of Section 2 for deriving the Coulomb branch relations, in the case of rank one. In Appendix C we show a method to extract the Coulomb branch relations in terms of a minimal basis of CB generators, namely we show explicitly how to integrate out the auxiliary generators, and apply it to the case of the $T_{USp(2N),2N}$ Coulomb branch. Appendices B and D gather other computations used in the main text.

## 2 From the abelianised relations to the polynomial relation

In this section we derive the CB relations for the $USp(2N)$ SQCD theory with $N_f$ flavours of fundamental hypermultiplets (or rather $2N_f$ half-hypermultiplets), with complex masses $m_{\alpha = 1, \cdots, N_f}$ associated to the flavour symmetry $O(2N_f)$. We employ the method developed in [12], starting from the "abelianised relations", which are CB relations for the monopole operators in the low-energy abelian theory at generic points of the Coulomb branch. Along the way we will slightly reformulate the proposal of [12], resolving certain ambiguities of the naive algorithm.

An algorithm for finding a minimal basis of generators and their CB relations is schematically as follows:

- Consider the full set of abelianised relations, involving abelian monopole operators of all magnetic charges in $\Lambda_{cochar}$, and solve for the abelian monopoles when possible. This leaves a finite set of abelian monopoles and abelian complex scalars, constrained by a set of abelianised relations which define the "abelianised CB variety". The abelianised

relations are valid in the complement of the discriminant locus $\Delta$, which is the location of non-abelian gauge symmetry enhancement.

- Extend the abelianised variety thus obtained to the discriminant locus $\Delta$, without introducing spurious branches.

- To obtain a description in terms of gauge invariant operators, quotient the extension of the abelianised variety by the action of the Weyl group. The resulting generators form a minimal basis of CB generators and the resulting relations are the CB relations of the non-abelian theory, valid on the full Coulomb branch, including possible loci of enhanced gauge symmetry.

The second step above involves some subtleties, that we will explain for the $USp(2N)$ SQCD theory.

The abelian theory at a generic point on the $USp(2N)$ Coulomb branch has gauge group $U(1)^N$, therefore the abelian chiral operators (in a chosen complex structure) are $N$ complex scalars $\varphi_a$ and abelian monopole operators $v_A$, labeled by a magnetic charge $A = (n_1, n_2, \cdots, n_N) \in \mathbb{Z}^N$. We will denote their vevs by the same notation, and use the convention $v_{(0,\cdots,0)} = 1$. The abelianised ring relations of [12] read

$$
v_A v_{\widetilde{A}} \prod_{a=1}^{N} (2\varphi_a)^{2(|n_a|+|\widetilde{n}_a|-|n_a+\widetilde{n}_a|)} \prod_{a<b} \prod_{s=\pm 1} (\varphi_a + s\varphi_b)^{|n_a+sn_b|+|\widetilde{n}_a+s\widetilde{n}_b|-|n_a+sn_b+\widetilde{n}_a+s\widetilde{n}_b|}
$$
$$
= v_{A+\widetilde{A}} \prod_{\alpha=1}^{N_f} \prod_{a=1}^{N} (\varphi_a^2 - m_\alpha^2)^{\frac{1}{2}(|n_a|+|\widetilde{n}_a|-|n_a+\widetilde{n}_a|)}, \tag{2.1}
$$

with magnetic charges $A = (\{n_a\})$, $\widetilde{A} = (\{\widetilde{n}_a\})$ and complex masses $m_\alpha$ for the fundamental hypermultiplets.[7] These abelianised relations hold only in the complement of the discriminant locus $\Delta = \{\exists a, \varphi_a = 0\} \cup \{\exists(a,b), a \neq b, \varphi_a^2 = \varphi_b^2\}$, which is the subvariety where the gauge symmetry enhances to a non-abelian symmetry.

A minimal basis of abelian operators is given by the monopoles of minimal charge $u_{a=1,\cdots,N}^{\pm}$, corresponding to monopole operators $v_A$ of magnetic charge $n_b = \pm\delta_{ab}$, as well as $\varphi_{a=1,\cdots,N}$. The other abelian monopoles can be expressed as polynomials in $u_a^{\pm}, \varphi_a$ and can therefore be eliminated from the description of the abelianised variety. The remaining abelianised relations are:

$$
u_a^+ u_a^- (2\varphi_a)^4 \prod_{a\neq b} (\varphi_a^2 - \varphi_b^2)^2 = \prod_{\alpha=1}^{N_f} (\varphi_a^2 - m_\alpha^2), \qquad a = 1, \cdots, N. \tag{2.2}
$$

As we have stressed, the abelianised relations (2.2) only hold in the complement of $\Delta$. To obtain the CB relations of the non-abelian theory one should extend the abelian relations to $\Delta$ and then perform the quotient by the Weyl group. We point out the subtlety that there are several ways to extend the relations to $\Delta$. For instance at zero masses $m_\alpha = 0$, with $N_f \geq 2$, we can simplify by a factor $\varphi_a^4$ on both sides of the abelianized relations. This does not affect the algebraic variety on the complement of $\Delta$, where $\varphi_a \neq 0$ for all $a$. However it does affect the extension to $\Delta$ by suppressing some branches emanating from the the locus $\Delta$. In addition we have the freedom to change the basis of abelian generators allowing mixing with the operators $\varphi_a^{-1}, (\varphi_a \pm \varphi_b)^{-1}$, which are well-defined on the complement of $\Delta$. Our prescription for extending the abelianized relations to $\Delta$ is to change the basis of abelian generators and simplify by common factors that appear on both sides of the abelianized relations (2.2),

---

[7]Compared to the conventions of [12], we have absorbed in the definition of monopole operators a possible sign depending on the magnetic charge.

whenever this is possible. The resulting abelianized relations should then be considered valid on the discriminant locus $\Delta$. This implies that we discard (some) branches emanating from $\Delta$, which we claim are spurious. We provide a justification of this prescription in appendix A by comparing our results with the hyperkähler quotient construction of the CB relations for $SU(2) \cong USp(2)$ SQCD theories.

We thus first massage the abelianized relations working in the complement of $\Delta$. The crucial step is to make the following change of variables for the abelian monopole operators, which is allowed outside $\Delta$,

$$\hat{u}_a^\pm = u_a^\pm - \frac{i^{N_f} \prod_\alpha m_\alpha}{(2\varphi_a)^2 \prod_{b \neq a}(\varphi_a^2 - \varphi_b^2)} \qquad (a = 1, \ldots, N) . \tag{2.3}$$

The abelianized relations (2.2) can then be massaged to

$$16\hat{u}_a^+ \hat{u}_a^- \varphi_a^2 \prod_{b \neq a}(\varphi_a^2 - \varphi_b^2)^2 + 4i^{N_f} \prod_{\alpha=1}^{N_f} m_\alpha \cdot (\hat{u}_a^+ + \hat{u}_a^-) \prod_{b \neq a}(\varphi_a^2 - \varphi_b^2) = \frac{P(\varphi_a^2) - P(0)}{\varphi_a^2} , \tag{2.4}$$

where $P(w) = \prod_{\alpha=1}^{N_f}(w - m_\alpha^2) := \sum_{n=0}^{N_f}(-)^n M_n w^{N_f - n}$, with $M_0 = 1$. Importantly, we divided (2.2) by $\varphi_a^2$, which is allowed outside $\Delta$. The right-hand-side of (2.4) is then a polynomial of degree $N_f - 1$ in $\varphi_a^2$:

$$\frac{P(\varphi_a^2) - P(0)}{\varphi_a^2} = \sum_{n=0}^{N_f-1}(-1)^n M_n \varphi_a^{2(N_f - 1 - n)} . \tag{2.5}$$

Note that $\prod_{\alpha=1}^{N_f} m_\alpha = \sqrt{M_{N_f}} = \mathrm{Pf}(m)$ is the pfaffian of the mass matrix. In (2.3) and (2.4) it is understood that $\prod_\alpha m_\alpha = \mathrm{Pf}(m)$ reduces to 1 for $N_f = 0$.

The relations (2.4) are deformations of the abelianized relations of the massless theory by lower order terms. We claim that it is this description that should be extended to the discriminant locus $\Delta$. Note that in the presence of a massless matter hypermultiplet the redefinition (2.3) trivializes.

The final step to obtain the CB relations is to perform the quotient by the Weyl group, which is generated by

$$\begin{aligned} \underline{\mathbb{Z}_2^{(a)}} : \quad & (\hat{u}_a^+, \hat{u}_a^-, \varphi_a) \to (\hat{u}_a^-, \hat{u}_a^+, -\varphi_a), \\ \underline{S_N} : \quad & \text{permutations of the triples } (\hat{u}_a^+, \hat{u}_a^-, \varphi_a). \end{aligned} \tag{2.6}$$

The Weyl invariant generators are efficiently packaged in the generating polynomials

$$\begin{aligned} Q(w) &= \prod_{a=1}^N (w - \varphi_a^2) := \sum_{n=0}^N (-)^n \Phi_n w^{N-n}, \quad \text{with } \Phi_0 = 1 , \\ U(w) &= \sum_{a=1}^N 2(\hat{u}_a^+ + \hat{u}_a^-) \prod_{b \neq a}(w - \varphi_b^2) = \sum_{n=0}^{N-1}(-)^n U_n w^{N-1-n} , \\ V(w) &= \sum_{a=1}^N 2(\hat{u}_a^+ - \hat{u}_a^-) \varphi_a \prod_{b \neq a}(w - \varphi_b^2) := \sum_{n=0}^{N-1}(-)^n V_n w^{N-1-n} . \end{aligned} \tag{2.7}$$

The non-abelian generators are then the coefficients $\Phi_n, U_n, V_n$ of the generating polynomials and correspond to the vevs of chiral operators in the non-abelian theory. The generators

$\Phi_n = \sum_{i_1 < \cdots < i_n} \varphi_{i_1}^2 \ldots \varphi_{i_n}^2$ are the elementary symmetric polynomials in $\varphi_a^2$.[8] The generators $U_n, V_n$ are the vevs of the monopole operators with minimal magnetic charges $(1, 0, \cdots, 0)$ but different dressing factors (see section 5.2 of [6] for details).

In the quotient variety, the abelian relations (2.4) become relations between the CB generators $\Phi_n, U_n, V_n$. They are conveniently packaged in the polynomial relation

$$wU(w)^2 + 2i^{N_f} \prod_\alpha m_\alpha \cdot U(w) - V^2(w) - \frac{P(w) - P(0)}{w} + Q(w)\widetilde{Q}(w) = 0 \quad \forall w \in \mathbb{C}, \quad (2.8)$$

where we introduced an auxiliary polynomial $\widetilde{Q}(w)$ of degree $\widetilde{N} \equiv \max(N_f - N, N) - 1$,

$$\widetilde{Q}(w) = \sum_{n=0}^{\widetilde{N}} (-)^n \widetilde{\Phi}_n w^{\widetilde{N} - n}. \quad (2.9)$$

The generators $\widetilde{\Phi}_{n=0, \cdots, \widetilde{N}}$ are auxiliary: they appear linearly in $\widetilde{N} + 1$ CB relations (see below), which can be used to solve for them, leading to a description purely in terms of $\Phi_n, U_n$ and $V_n$ generators, as in [6].[9] We will however find it convenient to keep the auxiliary generators $\widetilde{\Phi}_n$ in the description.

In the following we will set all the masses $m_\alpha$ to zero. The polynomial relation (2.8) then simplifies to

$$\begin{aligned} \underline{N_f = 0}: \quad & wU(w)^2 + 2U(w) - V(w)^2 + Q(w)\widetilde{Q}(w) = 0, \\ \underline{N_f \geq 1}: \quad & wU(w)^2 - V(w)^2 - w^{N_f - 1} + Q(w)\widetilde{Q}(w) = 0. \end{aligned} \quad (2.10)$$

Explicitly, the CB relations are

$$\underline{N_f = 0}: \quad [k = 0, \cdots, 2N - 1]$$

$$R_k := \sum_{n_1 + n_2 = k} U_{n_1} U_{n_2} + \sum_{n_1 + n_2 = k-1} V_{n_1} V_{n_2} + \sum_{n_1 + n_2 = k} \Phi_{n_1} \widetilde{\Phi}_{n_2} + 2(-)^N U_{k-N} = 0 \quad (2.11)$$

$$\underline{N_f \geq 1}: \quad [k = 0, \cdots, \widetilde{N} + N]$$

$$R_k :=$$

$$\sum_{\substack{n_1 + n_2 \\ = N - \widetilde{N} + k - 1}} U_{n_1} U_{n_2} + \sum_{\substack{n_1 + n_2 \\ = N - \widetilde{N} + k - 2}} V_{n_1} V_{n_2} - \sum_{\substack{n_1 + n_2 \\ = k}} (-)^{N + \widetilde{N}} \Phi_{n_1} \widetilde{\Phi}_{n_2} - (-)^{N_f} \delta_{k, \widetilde{N} + N - N_f + 1} = 0, \quad (2.12)$$

where it is understood that $U_{k-N} = 0$ if $k < N$. The auxiliary generators $\widetilde{\Phi}_k$ appear linearly in the relations $R_k$, $k = 0, \cdots, \widetilde{N}$, since $\Phi_0 = 1$.

For $N_f > 0$ and in absence of mass deformation, the Coulomb branch admits a $\mathbb{Z}_2$ global symmetry which acts non-trivially on the monopole operators,

$$\underline{\mathbb{Z}_2}: \quad U_n \rightarrow -U_n, \quad V_n \rightarrow -V_n. \quad (2.13)$$

It corresponds to a parity transformation inside the $O(2N_f)$ global symmetry which acts on the $2N_f$ half-hypermultiplets.[10]

---

[8] The ring of Casimir invariants is alternatively generated by the first $N$ power sums $p_k = \text{Tr}\,\varphi^{2k} = \sum_{a=1}^N \varphi_a^{2k}$, which are related to the elementary symmetric polynomials by Newton identities.

[9] The CB relations (2.8) may be written equivalently as

$$wU(w)^2 + 2i^{N_f} \prod_\alpha m_\alpha \cdot U(w) - V^2(w) - \frac{P(w) - P(0)}{w} = 0 \quad \mod \quad Q(w) = 0.$$

The abelianized relations (2.4) are recovered at the values $w = \varphi_a^2$.

[10] The $\mathbb{Z}_2$ parity acts on monopole operators due to fermionic matter zero modes in the monopole background (see footnote 5 of [1] and section 5.2.2 of [2]). It also acts on the Pfaffian of the mass matrix and the action on $U(w)$ is consistent with the mass deformed CB relations (2.8).

The CB relations also admit a $\mathbb{C}^*$ action (the complexification of the $U(1)_R$ symmetry) which acts on the generators with charges

$$
\begin{aligned}
R[\Phi_n] &= 2n, \quad R[\widetilde{\Phi}_n] = 2(N_f - N - \widetilde{N} - 1) + 2n, \\
R[U_n] &= N_f - 2N + 2n, \quad R[V_n] = N_f - 2N + 1 + 2n.
\end{aligned}
\tag{2.14}
$$

These charges are all bigger than 1/2 if and only if $N_f \geq 2N + 1$, corresponding to "good" theories in the classification of [2].[11] Since the R-charges of the generators are positive, in this case the Coulomb branch is algebraically a complex cone[12] and the theory is scale invariant around the origin (tip of the cone), defined by $\Phi_{n>0} = U_{n\geq0} = V_{n\geq0} = \widetilde{\Phi}_{n>0} = 0$ and $\widetilde{\Phi}_0 = 1$, where it is expected to flow to a SCFT, which we will call $T_{USp(2N),N_f}$ later. Once the auxiliary generators are eliminated, one is left with $3N$ Coulomb branch generators ($\Phi_n$, $U_n$ and $V_n$) subject to $N$ relations. The R-charges of generators and relations precisely agree with the results of [6] for the Hilbert series of the Coulomb branch. Note that, had we not divided by $\varphi_a^2$ to obtain the new abelianized relations (2.4), we would have deduced chiral ring relations with incorrect R-charges.

When $N_f \leq 2N$, some monopole operators have negative or vanishing R-charge, corresponding to "bad" theories in the classification of [2], and the infrared physics has not been studied, to the best of our knowledge. It is the main focus of this paper to elucidate the infrared behaviour and classify the infrared fixed points of the bad $USp(2N)$ theories, using the algebraic description of Coulomb branch (2.10), (2.11), (2.12), that we worked out in this section. To do so we propose to study the singular locus of the Coulomb branch, where the theory flows to interacting fixed points.

# 3 Singular structure of the Coulomb branch

In this section we analyse the singular structure of the Coulomb branch of $USp(2N)$ gauge theories with $N_f$ flavours. Physically, singularities occur when matter fields (and W-bosons) become massless, leading to the opening of a Higgs branch of the moduli space.[13] We then study the geometry near singular points, revealing the infrared effective theories and in particular the infrared fixed points that one can reach. In the next section we will study Higgs and mixed branches and provide a comprehensive picture of the moduli space of vacua.

## 3.1 Good theories

We first consider the theories with $N_f \geq 2N + 1$, i.e. the "good" theories. After absorbing $(-)^{N_f}$ factors in redefinitions of $U$ and $V$ generators, the CB relations are

$$
R_k := \sum_{\substack{n_1+n_2 \\ =2N-N_f+k}} U_{n_1} U_{n_2} + \sum_{\substack{n_1+n_2 \\ =2N-N_f+k-1}} V_{n_1} V_{n_2} + \sum_{\substack{n_1+n_2 \\ =k}} \Phi_{n_1} \widetilde{\Phi}_{n_2} - \delta_{k,0} = 0
\tag{3.1}
$$

for $k = 0, \cdots, N_f - 1$. To find the singular loci in the Coulomb branch, we compute the Jacobian matrix $J = (J_k^{\ i}) = (\partial R_k / \partial \mathcal{O}_i)$ by differentiating the CB relations with respect to the CB

---

[11]To be precise, $\widetilde{\Phi}_0$ has zero charge, but the relation $R_0 = 0$ determines $\widetilde{\Phi}_0 = 1$. More generally, the $\widetilde{\Phi}_n$ are determined by the relations $R_n = 0$ and can be ignored in the analysis of R-charges.

[12]The hyperkähler metric on the Coulomb branch, however, depends on the dimensionful Yang-Mills coupling $g$. The space only becomes a metric cone in the "infrared" limit $g \to \infty$.

[13]Our analysis will show that the only loci of enhanced gauge symmetry in the quantum corrected Coulomb branch are also loci where massless matter fields arise and therefore always roots of Higgs branches.

generators $\mathcal{O}_i = (\Phi_n | \widetilde{\Phi}_m | U_p | V_q)$,

$$J_k{}^i d\mathcal{O}_i = \sum_{\substack{n_1+n_2 \\ =2N-N_f+k}} 2U_{n_1} dU_{n_2} + \sum_{\substack{n_1+n_2 \\ =2N-N_f+k-1}} 2V_{n_1} dV_{n_2} + \sum_{\substack{n_1+n_2 \\ =k}} (\Phi_{n_1} d\widetilde{\Phi}_{n_2} + \widetilde{\Phi}_{n_2} d\Phi_{n_1}), \qquad (3.2)$$

with $0 \le k \le N_f - 1$. Explicitly, we find

$$J =$$

$$\begin{pmatrix}
0 & & & 1 & & & & & & & \\
\widetilde{\Phi}_0 & 0 & & \Phi_1 & 1 & & & & & & \\
\vdots & \ddots & & \vdots & \ddots & & & & & & \\
\widetilde{\Phi}_{N'-1} & & & \Phi_{N'} & & & U_0' & & & & \\
\widetilde{\Phi}_{N'} & & & \Phi_{N'+1} & & & U_1' & U_0' & & V_0' & \\
\vdots & & & \vdots & & & & & & & \\
\widetilde{\Phi}_{N-1} & \cdots & & \widetilde{\Phi}_0 & \Phi_N & & \vdots & \ddots & & \vdots & \ddots \\
\vdots & & & \vdots & & \ddots & & & & & \\
\widetilde{\Phi}_{\widetilde{N}-1} & \cdots & & \widetilde{\Phi}_{N'-2} & & \Phi_N \cdots \Phi_1 & 1 & U_{N-1}' \cdots U_0' & V_{N-2}' \cdots V_0' & 0 \\
\widetilde{\Phi}_{\widetilde{N}} & \cdots & & \widetilde{\Phi}_{N'-1} & & & \Phi_1 & 0 & U_1' & V_{N-1}' \cdots & V_0' \\
\ddots & & & \vdots & & & \vdots & \ddots & \vdots & \ddots & \vdots \\
& & \widetilde{\Phi}_{\widetilde{N}} & \widetilde{\Phi}_{\widetilde{N}-1} & & \Phi_N \Phi_{N-1} & & U_{N-1}' & V_{N-2}' \\
& & & \Phi_{\widetilde{N}} & & \Phi_N & & 0 & V_{N-1}'
\end{pmatrix} \qquad (3.3)$$

where we have only indicated non-zero entries and we used the notation $\widetilde{N} = N_f - N - 1$, $N' = N_f - 2N$, $U_n' = 2U_n$ and $V_n' = 2V_n$, for compactness.

The singular locus corresponds to the points where the above matrix has rank smaller than $N_f$ and which belong to the Coulomb branch, that is satisfying (3.1). We find that the singular locus $\mathcal{C}_{\text{sing}}^{(1)}$ is given by configurations where the last row of $J$ vanishes, namely $\Phi_N = \widetilde{\Phi}_{N_f-N-1} = V_{N-1} = 0$.[14] The CB relations then imply $U_{N-1} = 0$ and that the subvariety described by the remaining generators is isomorphic to the Coulomb branch $\mathcal{C}_{USp(2N-2),N_f-2}$ of the $USp(2N-2)$ theory with $N_f - 2$ flavours, which is also a good theory,

$$\mathcal{C}_{\text{sing}}^{(1)} = \{\Phi_N = \widetilde{\Phi}_{N_f-N-1} = U_{N-1} = V_{N-1} = 0\} \cap \mathcal{C} \cong \mathcal{C}_{USp(2N-2),N_f-2}. \qquad (3.4)$$

This corresponds to the locus with $\text{rank}(J) \le N_f - 1$. By the same analysis the singular locus $\mathcal{C}_{\text{sing}}^{(1)}$ contains itself a singular sublocus $\mathcal{C}_{\text{sing}}^{(2)}$ where the operators $\Phi_{N-1} = \widetilde{\Phi}_{N_f-N-2} = U_{N-2} = V_{N-2} = 0$ vanish and corresponding to the subvariety of the Coulomb branch where $\text{rank}(J) \le N_f - 2$. Iterating the reasoning we find a nested sequence of singular subloci $\mathcal{C}_{\text{sing}}^{(r)}$ defined by the condition $\text{rank}(J) \le N_f - r$, with $1 \le r \le N$,

$$\mathcal{C}^* := \mathcal{C}_{\text{sing}}^{(N)} \subset \cdots \subset \mathcal{C}_{\text{sing}}^{(r)} \subset \cdots \subset \mathcal{C}_{\text{sing}}^{(1)} \subset \mathcal{C}_{\text{sing}}^{(0)} := \mathcal{C},$$

$$\begin{aligned}
\mathcal{C}_{\text{sing}}^{(r)} &= \{p \in \mathcal{C} \,|\, \text{rank}(J(p)) \le r\} \\
&= \{\Phi_{N+1-i} = \widetilde{\Phi}_{N_f-N-i} = U_{N-i} = V_{N-i} = 0 \,|\, i = 1, \cdots, r\} \cap \mathcal{C} \\
&\cong \mathcal{C}_{USp(2N-2r),N_f-2r}.
\end{aligned} \qquad (3.5)$$

---

[14] Strictly speaking we find that this locus is part of the singular locus, but we were not able to prove in general that there is no other singular locus. This locus can be associated to the vanishing of a pair $(u_a^+ + u_a^-, \varphi_a)$ for a given $a$, which makes some matter fields massless. This is therefore the root of a Higgs branch. We do not expect other singular loci on physical grounds, and for low $N$ we were able to confirm this expectation explicitly. The same comment applies to other singular locus computations in this paper.

The subvariety $\mathcal{C}_{\text{sing}}^{(r)}$ has quaternionic codimension $r$ inside $\mathcal{C}$. We will call it the *codimension $r$ singular locus*. The nested sequence of singular subvarieties ends at $r = N$ and the most singular locus $\mathcal{C}^*$ is a single point, described by $\Phi_{n>0} = U_{n\geq0} = V_{n\geq0} = \widetilde{\Phi}_{n>0} = 0$ and $\widetilde{\Phi}_0 = 1$. This is the origin of the Coulomb branch, where the theory flows to an SCFT, which we call $T_{USp(2N),N_f}$.

## 3.2 Bad theories

We now consider bad $USp(2N)$ SQCD theories, which have $0 \leq N_f \leq 2N$ flavours. For the sake of presentation, we will start from the simplest case of the pure SYM theory with no flavours, and then increase the number of flavours up to $2N$.

For $N_f = 0$ the CB relations are given by (2.11), which we reproduce here for convenience:

$$R_k := \sum_{n_1+n_2=k} U_{n_1} U_{n_2} + \sum_{n_1+n_2=k-1} V_{n_1} V_{n_2} + \sum_{n_1+n_2=k} \Phi_{n_1} \widetilde{\Phi}_{n_2} + 2(-)^N U_{k-N} = 0 , \qquad (3.6)$$

for $k = 0, \cdots, 2N-1$. The Jacobian matrix is

$$J =$$

$$\left(\begin{array}{cccc|cccc|cccc|cccc}
0 & & & & 1 & & & & U_0' & & & & 0 & & & \\
\widetilde{\Phi}_0 & 0 & & & \Phi_1 & 1 & & & U_1' & U_0' & & & V_0' & 0 & & \\
\vdots & \ddots & \ddots & & \vdots & & \ddots & & \vdots & & \ddots & & \vdots & & \ddots & \\
\widetilde{\Phi}_{N-2} & \cdots & \widetilde{\Phi}_0 & 0 & \Phi_{N-1} & \cdots & \Phi_1 & 1 & U_{N-1}' & \cdots & & U_0' & V_{N-2}' & \cdots & V_0' & 0 \\
\widetilde{\Phi}_{N-1} & & \cdots & \widetilde{\Phi}_0 & \Phi_N & \cdots & & \Phi_1 & c_N & U_{N-1}' & \ddots & U_1' & V_{N-1}' & \cdots & & V_0' \\
& \ddots & & \vdots & & \ddots & & \vdots & & \ddots & & \vdots & & \ddots & & \vdots \\
& & \widetilde{\Phi}_{N-1} & \widetilde{\Phi}_{N-2} & & & \Phi_N & \Phi_{N-1} & & & c_N & U_{N-1}' & & & V_{N-1}' & V_{N-2}' \\
& & & \widetilde{\Phi}_{N-1} & & & & \Phi_N & & & & c_N & & & & V_{N-1}'
\end{array}\right)$$
$$(3.7)$$

where again $U_n' = 2U_n$, $V_n' = 2V_n$, and we defined $c_N = 2(-)^N$. The locus where the above Jacobian matrix has reduced rank does not intersect the Coulomb branch, therefore the Coulomb branch of the $USp(2N)$ theory with $N_f = 0$ is smooth.

For $1 \leq N_f \leq 2N$, the CB relations are

$$R_k := \sum_{n_1+n_2=k} (U_{n_1} U_{n_2} + \Phi_{n_1} \widetilde{\Phi}_{n_2}) + \sum_{n_1+n_2=k-1} V_{n_1} V_{n_2} - (-)^{N_f} \delta_{k,2N-N_f} = 0, \qquad (3.8)$$

for $k = 0, \cdots, 2N-1$. The Jacobian matrix of this system of equations is given by

$$J =$$

$$\left(\begin{array}{cccc|cccc|cccc|cccc}
0 & & & & 1 & & & & U_0' & & & & 0 & & & \\
\widetilde{\Phi}_0 & 0 & & & \Phi_1 & 1 & & & U_1' & U_0' & & & V_0' & 0 & & \\
\vdots & \ddots & \ddots & & \vdots & & \ddots & & \vdots & & \ddots & & \vdots & & \ddots & \\
\widetilde{\Phi}_{N-2} & \cdots & \widetilde{\Phi}_0 & 0 & \Phi_{N-1} & \cdots & \Phi_1 & 1 & U_{N-1}' & \cdots & & U_0' & V_{N-2}' & \cdots & V_0' & 0 \\
\widetilde{\Phi}_{N-1} & & \cdots & \widetilde{\Phi}_0 & \Phi_N & \cdots & & \Phi_1 & 0 & U_{N-1}' & \ddots & U_1' & V_{N-1}' & \cdots & & V_0' \\
& \ddots & & \vdots & & \ddots & & \vdots & & \ddots & & \vdots & & \ddots & & \vdots \\
& & \widetilde{\Phi}_{N-1} & \widetilde{\Phi}_{N-2} & & & \Phi_N & \Phi_{N-1} & & & 0 & U_{N-1}' & & & V_{N-1}' & V_{N-2}' \\
& & & \widetilde{\Phi}_{N-1} & & & & \Phi_N & & & & 0 & & & & V_{N-1}'
\end{array}\right)$$
$$(3.9)$$

which has reduced rank when $\Phi_N = \widetilde{\Phi}_{N-1} = V_{N-1} = 0$. Depending on the value of $N_f$ this locus intersects the Coulomb branch in different ways.

For $N_f = 1$ this locus does not intersect the Coulomb branch, since it violates the CB relation $\Phi_N \widetilde{\Phi}_{N-1} + V_{N-1}^2 = -1$.

For $N_f = 2$, this locus intersects the Coulomb branch. The CB relation at $k = 2N - 2$ then requires $U_{N-1}^2 = 1$, with two solutions $U_{N-1,\pm} = \pm 1$. The remaining CB relations combine into the polynomial relation

$$w U'(w)^2 + 2U'(w) - V'(w)^2 + Q'(w)\widetilde{Q}'(w) = 0, \tag{3.10}$$

where $U'(w) = \pm(-)^{N-1} w^{-1}(U(w) - U(0)) = w^{-1}(\pm(-)^{N-1} U(w) - 1)$, $V'(w) = w^{-1} V(w)$, $\widetilde{Q}'(w) = w^{-1}\widetilde{Q}(w)$ and $Q'(w) = w^{-1}Q(w)$ are polynomials of degrees $N-2, N-2, N-2$ and $N-1$ respectively. This is the polynomial relation describing the Coulomb branch of the pure $USp(2N-2)$ SYM theory $\mathcal{C}_{USp(2N-2),0}$. Therefore we find that the singular locus has two disjoint components each isomorphic to $\mathcal{C}_{USp(2N-2),0}$.

For $N_f \geq 3$, the CB relations imply the further condition $U_{N-1} = 0$ and reduce to the CB relations of the $USp(2N-2)$ theory with $N_f - 2$ flavours, which is also a bad theory. Therefore the situation is similar to what we found for good theories: the Coulomb branch has a nested sequence of singular subloci of increasing (quaternionic) codimension $r$, defined by

$$\begin{aligned}
\mathcal{C}_{\text{sing}}^{(r)} &= \{\Phi_{N+1-i} = \widetilde{\Phi}_{N-i} = U_{N-i} = V_{N-i} = 0 \,|\, i = 1, \cdots, r\} \cap \mathcal{C} \\
&\cong \mathcal{C}_{USp(2N-2r),N_f-2r}.
\end{aligned} \tag{3.11}$$

(This holds for $r < N_f/2$. The case $r = N_f/2$ for $N_f$ even is special and is discussed below.) For odd $N_f$ the sequence terminates at $r = \frac{N_f-1}{2}$ and the most singular locus is isomorphic to the smooth Coulomb branch of the $USp(2N-N_f+1)$ theory with one flavour $\mathcal{C}_{USp(2N-N_f+1),1}$,

$$\begin{aligned}
\underline{N_f \text{ odd}:} \quad &\mathcal{C}^* \equiv \mathcal{C}_{\text{sing}}^{((N_f-1)/2)} \subset \cdots \subset \mathcal{C}_{\text{sing}}^{(1)} \subset \mathcal{C}_{\text{sing}}^{(0)} \equiv \mathcal{C}, \\
&\mathcal{C}_{\text{sing}}^{((N_f-1)/2)} \cong \mathcal{C}_{USp(2N-N_f+1),1}.
\end{aligned} \tag{3.12}$$

For even $N_f$, the sequence stops at the codimension $r = N_f/2 - 1$ singular locus, which is isomorphic to the Coulomb branch of the $USp(2N - N_f + 2)$ theory with two flavours $\mathcal{C}_{USp(2N-N_f+2),2}$, and whose singular locus consists of two disjoint copies of the smooth Coulomb branch $\mathcal{C}_{USp(2N-N_f),0}$ as explained above. Therefore the sequence of nested singularities has the form

$$\begin{aligned}
\underline{N_f \text{ even}:} \quad &\mathcal{C}^* \equiv \mathcal{C}_{\text{sing}+}^{(N_f/2)} \cup \mathcal{C}_{\text{sing}-}^{(N_f/2)} \subset \mathcal{C}_{\text{sing}}^{(N_f/2-1)} \subset \cdots \subset \mathcal{C}_{\text{sing}}^{(1)} \subset \mathcal{C}_{\text{sing}}^{(0)} \equiv \mathcal{C}, \\
&\mathcal{C}_{\text{sing}\pm}^{(N_f/2)} \cong \mathcal{C}_{USp(2N-N_f),0}.
\end{aligned} \tag{3.13}$$

When $N_f = 2N$, $\mathcal{C}^*$ consists of two points only. We will call the two subvarieties $\mathcal{C}_{\text{sing}\pm}^{(N_f/2)}$ special singular loci and we will use the notation $\mathcal{C}^{*\pm} \equiv \mathcal{C}_{\text{sing}\pm}^{(N_f/2)}$. In terms of generators, the special singular loci are described by

$$\mathcal{C}^{*\pm} = \{\Phi_{N+1-i} = \widetilde{\Phi}_{N-i} = V_{N-i} = 0, \, U_{N-i} = \pm\delta_{i,N_f/2} \,|\, i = 1, \cdots, \tfrac{N_f}{2}\} \cap \mathcal{C}. \tag{3.14}$$

The most singular locus $\mathcal{C}^*$ in the Coulomb branch has therefore either a single component, when $N_f$ is odd, or two disjoint components, when $N_f$ is even and positive. These components have positive dimension, except for the $N_f = 2N$ theory. As we will see in later sections, the most singular locus $\mathcal{C}^*$ is the root of the Higgs branch $\mathcal{H}$ of the theory. The fact that $\mathcal{C}^*$ has a

positive dimension means that the theory cannot be completely Higgsed for $N_f < 2N$, so that the maximally Higgsed component of the moduli space of vacua is of the form $\mathcal{C}^* \times \mathcal{H}$. We will comment later on the interpretation when $\mathcal{C}^*$ has two separate components.

To summarize, we find qualitative differences between the Coulomb branch of vacua of $USp(2N)$ gauge theories with $N_f$ flavours for different values of $N_f$ and $N$. For good theories ($N_f > 2N$) the Coulomb branch is an algebraic cone and the most singular locus $\mathcal{C}^*$ is a single point (the tip of the cone). For bad theories ($N_f \leq 2N$) $\mathcal{C}^*$ can have zero, one or two components. For $N_f \in \{0, 1\}$ the Coulomb branch is smooth and $\mathcal{C}^*$ is empty. For $2 < N_f < 2N$ odd, $\mathcal{C}^*$ has a single component of positive quaternionic dimension. For $2 \leq N_f \leq 2N$ even, $\mathcal{C}^*$ has two separate components with positive dimension, except for $N_f = 2N$, where they are two isolated points. These qualitative behaviours are schematically depicted in Figure 1.

## 3.3 Effective theory near the singular loci

In the previous subsections we have identified the nested sequence of singular subvarieties of the Coulomb branch for good and for bad theories. Next, we wish to understand the low-energy physics at each point on the Coulomb branch, and in particular at the singular points which signal the presence of extra massless states leading to an interacting SCFT. Information about the low-energy physics at a given point of the Coulomb branch of vacua is encoded in the local CB geometry near that point. Sometimes one recognizes the local geometry as the Coulomb branch of a known SCFT, indicating that the theory flows to that SCFT at low energies. We will see that most (but not all) of the low-energy fixed points can be understood in that way. We will later complete this analysis by studying the Higgs branches which emanate from the singular loci of the Coulomb branch.

At a generic point $p$ in the Coulomb branch it is well known that the low-energy theory is described by $N$ free twisted hypermultiplets. In our algebro-geometric language, this can be derived from the CB relations by expanding every generator into its VEV at $p$ plus a small fluctuation $\mathcal{O}_i = \mathcal{O}_{i,0}(p) + \delta\mathcal{O}_i$. For generic (non-zero) VEVs $\mathcal{O}_{i,0}(p)$, the $\max(N_f, 2N)$ CB relations become linear relations for the $\delta\mathcal{O}_i$ and can be used to solve for as many generators, leaving $2N$ remaining unconstrained generators. These unconstrained generators parametrize the local flat geometry $\mathbb{C}^{2N}$ near a smooth point and are identified with the VEVs of $N$ free twisted hypermultiplets.

The analysis of the infrared physics near the non-special singular loci $\mathcal{C}_{\text{sing}}^{(r)}$ proceeds in the same way for good and bad theories, the only difference being the allowed range of $r$. To study the geometry near a singular locus one should take certain limits, where the VEVs of some generators are small, in the CB relations and rewrite the resulting CB relations as those of a known theory. In practice this is not always straightforward to do. Fortunately the results can be found more easily from the abelianised relations. The codimension $r$ singular locus $\mathcal{C}_{\text{sing}}^{(r)}$ is characterized by having the operators $\Phi_{N+1-i}, \widetilde{\Phi}_{\widetilde{N}-i}, U_{N-i}, V_{N-i}$ vanishing for $i = 1, \cdots, r$ (see (3.5)). This implies that, up to gauge transformations, we are looking at a vacuum where $r$ pairs $(u_a^+ + u_a^-, \varphi_a)_{a=1,\cdots,r}$ vanish in the abelianised variety. The abelianised relations (2.4) (at zero masses) for $\varphi_{a=1,\cdots,r}$ small compared to $\varphi_{a=r+1,\cdots,N}$ reduce to

$$16 u_a^+ u_a^- \varphi_a^2 \prod_{\substack{b=1 \\ b \neq a}}^{r} (\varphi_a^2 - \varphi_b^2)^2 \prod_{c=r+1}^{N} \varphi_c^4 = \varphi_a^{2N_f - 2}, \quad a = 1, \cdots, r,$$

$$16 u_a^+ u_a^- \varphi_a^2 \prod_{\substack{b=r+1 \\ b \neq a}}^{N} (\varphi_a^2 - \varphi_b^2)^2 = \varphi_a^{2(N_f - 2r) - 2}, \quad a = r+1, \cdots, N.$$

$$(3.15)$$

The relations in the first line can be interpreted as the abelianised relations of a $USp(2r)$ theory

with $N_f$ flavours, upon absorbing the (non-zero) factor $\prod_{c=r+1}^{N} \varphi_c^4$ into redefinitions of the $u_a^\pm$. After the quotient by the Weyl group this describes a variety isomorphic to $\mathcal{C}_{USp(2r),N_f}$, the Coulomb branch of the good $USp(2r)$ theory with $N_f$ flavours, or equivalently the Coulomb branch of the $T_{USp(2r),N_f}$ SCFT, that we are probing at its origin. On the other hand, the relations inn the second line match the abelianised relations of the $USp(2N-2r)$ theory with $N_f - 2r$ behaviours and we are zooming near a generic point in the abelianised variety (since we take the scalars $\varphi_{a=r+1,\cdots,N}$ generic). The local geometry near such a generic point is simply $\mathbb{C}^{2(N-r)}$, parametrised by $N-r$ free twisted hypermultiplets.

We conclude that the local Coulomb branch geometry $\mathcal{U}[\mathcal{C}_{\text{sing}}^{(r)}]$ near a generic point $p$ of the codimension $r$ singular locus $\mathcal{C}_{\text{sing}}^{(r)}$ is

$$\mathcal{U}[\mathcal{C}_{\text{sing}}^{(r)}] \cong \mathbb{C}^{2(N-r)} \times \mathcal{C}_{USp(2r),N_f}, \tag{3.16}$$

and that the infrared theory has an interacting fixed point $T_{USp(2r),N_f}$ and $N-r$ free twisted hypermultiplets

$$p \in \mathcal{C}_{\text{sing}}^{(r)} \xrightarrow{IR} T_{USp(2r),N_f} + (N-r) \text{ free twisted hypers}. \tag{3.17}$$

This is valid for all $r = 0, 1, \cdots, N$ for good theories. For bad theories, this holds for $r = 0, 1, \cdots, \lfloor \frac{N_f-1}{2} \rfloor$. The only singular loci left to discuss are the two special singular loci $\mathcal{C}^{*\pm} \equiv \mathcal{C}_{\text{sing}\pm}^{(N_f/2)}$, which are the most singular loci of bad theories with even $N_f$. Note that all the $T_{USp(2r),n_f}$ SCFTs appearing so far obey $n_f > 2r$, as they should by definition. This means that the fixed points of bad theories discussed so far are identified with fixed points (sitting at the origin of the CB of) good theories. This is analogous to the situation of $U(N)$ SQCD theories described in [14], where all the fixed points were found to be in the class $T_{U(r),n_f}$ with $n_f \geq 2r$. We will see however that for $USp(2N)$ theories there *are* new fixed points which arise at the special singular loci $\mathcal{C}^{*\pm}$. Studying the local geometry near vacua of $\mathcal{C}^{*\pm}$ requires a slightly longer and richer discussion that we present in the next subsection.

## 3.4 SCFT at the special singular loci

The local geometries close to the special singular subvarieties $\mathcal{C}^{*+}$ and $\mathcal{C}^{*-}$ of the bad $USp(2N)$ theory with even $N_f$ are identical: they are mapped to each other by the $\mathbb{Z}_2$ symmetry (2.13) acting on monopole operators, so we will discuss $\mathcal{C}^{*+}$ only.

At the level of the abelianized relations one reaches a point in $\mathcal{C}^{*+}$ described by (3.14) by letting the pairs $(u_a^+ + u_a^-, \varphi_a)_{a=1,\cdots,N_f/2-1}$ go to zero and the pair $(u_{N_f/2}^+ + u_{N_f/2}^-, \varphi_{N_f/2})$ go to $(\frac{1}{2} \prod_{b=N_f/2+1}^{N} \varphi_b^{-2}, 0)$, whereas the abelianized variables with $a = N_f/2 + 1, \cdots, N$ are of order 1. Using the same reasoning as above, we find that the local geometry has a factor $\mathcal{U}[\mathcal{C}_{USp(N_f),N_f}^{*+}]$ isomorphic to the geometry near a special singular point of the Coulomb branch $\mathcal{C}_{USp(N_f),N_f}$[15], and a factor $\mathbb{C}^{2N-N_f}$, which is nothing but the tangent space at any point along $\mathcal{C}^{*+} \cong \mathbb{C}^{2N-N_f}$,

$$\mathcal{U}[\mathcal{C}^{*\pm}] \cong \mathbb{C}^{2N-N_f} \times \mathcal{U}[\mathcal{C}_{USp(N_f),N_f}^{*\pm}]. \tag{3.18}$$

The problem is then reduced to studying the local geometry around a special singular point in the theory with $N_f = 2N$. We accomplish this task in appendix B by studying directly the local geometry in the CB relations written in terms of gauge invariant operators. The analysis involves a redefinition of the generators $(U_n, \Phi_n) \to (U_n', \Phi_n')$ which is only valid in a

---

[15]Indeed the Coulomb branch of the $USp(N_f)$ theory with (even) $N_f$ flavours has special singular loci which are points.

neighborhood of the special vacuum and which makes manifest the emergence of a new $U(1)$ global symmetry.

The final CB relations (B.9) read

$$\underline{k = 1, \cdots, 2N-1}: \quad R_k := \sum_{n_1+n_2=k} U'_{n_1} U'_{n_2} + \sum_{n_1+n_2=k-1} (\Phi'_{n_1} \widetilde{\Phi}_{n_2} + V_{n_1} V_{n_2}) = 0, \quad (3.19)$$

with generators $U'_{n=1,\cdots,N-1}, \Phi'_{n=0,\cdots,N-1}, \widetilde{\Phi}_{n=0,\cdots,N-1}, V_{n=0,\cdots,N-1}$ and constant $U'_0 = 1$.

In this reformulation an accidental $U(1)_J$ global symmetry is manifest, which rotates the generators $\Phi'_n$ and $\widetilde{\Phi}_n$ with opposite charges $\pm 1$. The IR $U(1)_R$ symmetry acting on these generator is a combination of the UV R-symmetry[16] and this $U(1)_J$ accidental symmetry leading to the IR R-charges

$$R[\Phi'_n]_{n=0,\cdots,N-1} = 2n+1, \quad R[\widetilde{\Phi}_n]_{n=0,\cdots,N-1} = 2n+1,$$
$$R[U'_n]_{n=1,\cdots,N-1} = 2n, \quad R[V_n]_{n=0,\cdots,N-1} = 2n+1. \quad (3.20)$$

All R-charges are bigger than or equal to one, as appropriate for an interacting SCFT.

The relations $R_k = 0$ in (3.19) have R-charges $2k$, with $k = 1, \ldots, 2N-1$. The first $N-1$ relations (with $k = 1, \ldots, N-1$) can be used to eliminate the auxiliary generators $U'_n$, $1 \le n \le N-1$, which appear linearly (since $U'_0 = 1$). One is then left with $N$ remaining nontrivial relations with R-charges equal to $2N, 2(N+1), \ldots, 2(2N-1)$.

This furnishes a new class of interacting fixed points $T_{USp(2N),2N}$ which cannot be reached in good theories (except for the $N = 1$ case as explained below). Instead, the $T_{USp(2N),2N}$ CFT can be found at either of the two special singular points of the Coulomb branch of the $USp(2N)$ theory with $2N$ flavours.

For $N = 1$ we have a single relation

$$\Phi'_0 \widetilde{\Phi}_0 + V_0^2 = 0, \quad (3.21)$$

corresponding to an $A_1$ singularity. This reproduces the results of [1] which studied the Coulomb branch of the $SU(2) \cong USp(2)$ theory with two flavours and found two isolated $A_1$ singularities. The $A_1$ singularity is nothing but the Coulomb branch of a $U(1)$ theory with two flavour hypermultiplets, which is a good theory, and the fixed point is the $T(SU(2))$ SCFT.

The CB relations (3.19) can be recast as a generating polynomial relation. Introducing the generating polynomials

$$U'(w) = \sum_{n=0}^{N-1} (-)^n U'_n w^{N-1-n}, \quad (U'_0 = 1)$$
$$Q'(w) = \sum_{n=0}^{N-1} (-)^n \Phi'_n w^{N-1-n}, \quad (3.22)$$

we find that for all complex values of $w$

$$w U'(w)^2 - Q'(w)\widetilde{Q}(w) - V(w)^2 = w^{2N-1}. \quad (3.23)$$

In this description the "auxiliary" polynomial is $U'(w)$, since the $U'_n$ generators appear linearly in certain relations and can be eliminated from the CB description, if needed. Although very similar to (2.10) (with $N_f = 2N$) the polynomial relation (3.23) differs from it in definitions of the generating polynomials that appear. We describe in appendix C how to solve for the

---

[16]The special vacua indeed preserve the UV $U(1)_R$ symmetry. Following the generator redefinitions in appendix B, the UV $U(1)_R$ charges of $U'_n$ and $\Phi'_n$ are respectively $2n$ and $2n+2$.

auxiliary generators $U'_n$ and provide a presentation of the CB relations in terms of the $\Phi'_n$, $V_n$ and $\widetilde{\Phi}_n$ generators only. The method presented there can be applied to solve for auxiliary generators $\widetilde{\Phi}_n$ in CB relations of the bad and good $USp(2N)$ theories as well.

From the study of the Higgs branch of the $USp(2N)$ theory with $2N$ flavours we will find that the $T_{USp(2N),2N}$ SCFT has an $SO(4N)$ flavour symmetry acting on its Higgs branch (except for $N = 1$). This implies that the SCFT admits $2N$ complex mass deformations which deform its Coulomb branch. If we reproduce the same steps that led to the CB of $T_{USp(2N),2N}$ starting from the mass deformed relations (2.8), with complex masses $m_\alpha$, $\alpha = 1, \cdots, 2N$, we end up with the deformed polynomial relation

$$wU'(w)^2 + 2(-1)^N \text{Pf}(m)U'(w) - Q'(w)\widetilde{Q}(w) - V(w)^2 = \frac{P(w) - P(0)}{w}, \qquad (3.24)$$

where we absorbed a term by a redefinition of $Q'(w)$, $Q'(w) + (-1)^N \text{Pf}(m) \to Q'(w)$. Equation (3.24) should therefore describe the Coulomb branch of the mass-deformed SCFT. Notice that the $SU(2)$ symmetry is preserved by the mass deformation.

Several important comments are in order. First, the CB relations of the $T_{USp(2N),2N}$ CFTs exhibit an accidental $U(1)_J$ flavour symmetry, which is actually part of an $SU(2)_J$ accidental symmetry. Indeed, in $\mathcal{N} = 4$ supersymmetry the R-symmetry acting on the Coulomb branch is $SU(2)_C$ and a mixing such as the one described above only makes sense if we have an $SU(2)_J$ symmetry. It is easy to see that CB relations are invariant under $SU(2)_J$ with $(\Phi'_n, V_n, \widetilde{\Phi}_n)$ being triplets and $U'_n$ being singlets. The presence of an $SU(2)_J$ triplet of chiral operators $(\Phi'_0, V_0, \widetilde{\Phi}_0)$ at R-charge one implies the existence of the three conserved currents of $SU(2)_J$ by superconformal symmetry [2,5]. The relation between the (unbroken) $SU(2)_C^{\text{UV}}$ and $SU(2)_C^{\text{IR}}$ is then

$$SU(2)_C^{\text{UV}} = \text{diag}(SU(2)_C^{\text{IR}} \times SU(2)_J). \qquad (3.25)$$

Secondly, we notice that the nontrivial generators of the $N$-quaternionic dimensional Coulomb branch of $T_{USp(2N),2N}$ have all odd integer R-charges/dimensions $1, 3, \ldots, 2N - 1$, and form triplets of an $SU(2)_J$ symmetry as explained. For $N > 1$ this cannot be achieved as the Coulomb branch of a good rank $N$ UV gauge theory. Indeed, the generators of the latter contain the $N$ Casimir invariants of the gauge group, which include the quadratic Casimir (of $R$-charge 2) for non-abelian simple factors. The absence of generators of $R$-charge 2 for $N > 1$ rules out a non-abelian gauge group. This leaves the possibility of an abelian $U(1)^N$ gauge group for $N = 1, 2, 3$. This would however lead to a $U(1)^N$ topological symmetry, which cannot be a subgroup of the full $SU(2)_J$ symmetry group acting on the CB of $T_{USp(2N),2N}$ if $N = 2, 3$, which are then ruled out. The only remaining option for a good UV gauge theory description is the gauge group $U(1)$ for $N = 1$, which is indeed realized [1] as we reviewed above. One may therefore think of the $T_{USp(2N),2N}$ SCFTs for $N > 1$ as being non-Lagrangian. It is a remarkable phenomenon that we can reach such fixed points on the moduli space of vacua of standard gauge theories. This is analogous to the original Argyres-Douglas fixed point [23] found on the Coulomb branch of a 4d $\mathcal{N} = 2$ gauge theory.

This completes our analysis of the infrared effective theory at various points on the Coulomb branch for bad theories $USp(2N)$ with even $N_f \leq 2N$ number of flavours. At a generic point of one of the special singular loci $\mathcal{C}^{*\pm}$, the infrared theory has an interacting fixed point $T_{USp(N_f),N_f}$ and $N - \frac{N_f}{2}$ free twisted hypermultiplets,

$$p \in \mathcal{C}^{*\pm} \xrightarrow{IR} T_{USp(N_f),N_f} + \left(N - \frac{N_f}{2}\right) \text{free twisted hypers}. \qquad (3.26)$$

In the following section we discuss the Higgs branch factors that emanate from the singular loci of the Coulomb branch and provide a fully consistent picture of the moduli space of vacua in the quantum theory.

Our results about the effective low energy theory at vacua belonging to the most singular locus $\mathcal{C}^*$, for the possible ranges of $N$ and $N_f \geq 2$, are summarized in Figure 1, where we indicated the SCFTs and the possible free twisted hypermultiplets. All *the nontrivial* SCFTs are of the form $T_{USp(2r),N_f}$ with $N_f > 2r$, or $T_{USp(2r),2r}$, namely there is no new interacting fixed point associated to the bad theories with $N_f < 2N$.

# 4 Higgs branch and full moduli space

It is well known that the Higgs branch of 3d $\mathcal{N} = 4$ gauge theories is classically exact [15] and can be described as a hyperKähler quotient [27]. If there are no continuous Fayet-Iliopoulos parameters, as is the case for gauge group $USp(2N)$, the Higgs branch is a hyperKähler cone. In fact the Higgs branch might consist of several components, which classically intersect each other along common subvarieties containing the origin of the Higgs branch (the *tip* of the cone). Pure Higgs branch components, where the gauge group is completely broken, intersect the Coulomb branch at a single point (the *root* of the Higgs branch component). Components of the Higgs branch where the gauge group is not completely broken are instead part of a mixed Higgs-Coulomb branch where some vector multiplet scalars also take vev, and their root is a nontrivial singular subvariety of the Coulomb branch. While quantum corrections do not affect the Higgs branch metric, they do affect the Coulomb branch and hence the location of the roots of the various Higgs branch components on the Coulomb branch. In this section we will first review known results on the Higgs branch of $USp(2N)$ SQCD theories [9,25,28], with minor corrections. By combining these known results with our new analysis of the Coulomb branch, we will then determine how the various components of the Higgs branch intersect the Coulomb branch at its singular loci, providing a complete picture of the quantum corrected moduli space of supersymmetric vacua.

## 4.1 Higgs branch

The matter content of $USp(2N)$ SQCD with $N_f$ massless flavours consists of half-hypermultiplets $Q = (Q^A{}_i)$, with $A = 1, \ldots, 2N$ and $i = 1, \ldots, 2N_f$, which transform in the bifundamental representation of the gauge symmetry $USp(2N)$ and the flavour symmetry $O(2N_f) = SO(2N_f) \rtimes \mathbb{Z}_2$. In a complex structure where the scalars $Q$ in the half-hypermultiplets are holomorphic, the Higgs branch of $USp(2N)$ SQCD with $N_f$ massless flavours can be described as a complex algebraic variety as

$$
\begin{aligned}
\mathcal{H} &= \{Q \in \mathbb{C}^{2N \times 2N_f} \mid QQ^T = 0\}/USp(2N)_{\mathbb{C}} \\
&\cong \{M \equiv Q^T J Q = -M^T \in \mathbb{C}^{2N_f \times 2N_f} \mid M^2 = 0, \ \mathrm{rk}(M) \leq 2\min(N, \lfloor \tfrac{N_f}{2} \rfloor)\} \,,
\end{aligned}
\tag{4.1}
$$

where $\mathbb{C}^{a \times b}$ denotes the space of $a$-by-$b$ complex matrices. The first line of (4.1) expresses the Higgs branch in terms of the gauge variant squarks $Q = (Q^A{}_i)$, subject to the $F$-term equation $QQ^T = 0$ and modded out by the action of the complexified gauge group (this is nothing but the Kähler quotient $\mathbb{C}^{2N \times 2N_f} // USp(2N)_{\mathbb{C}}$). The second line describes the Higgs branch in terms of gauge invariant operators, which are encoded in the size $2N_f$ antisymmetric meson matrix $M = (M_{ij}) = (J_{AB} Q^A{}_i Q^B{}_j)$ that is obtained by contracting the color indices of two squarks with the symplectic matrix

$$
J = (J_{AB}) \equiv \begin{pmatrix} 0 & 1 \\ -1 & 0 \end{pmatrix} \otimes \mathbb{1}_{N \times N} \,.
\tag{4.2}
$$

The meson matrix is antisymmetric, squares to zero because of the $F$-terms $QQ^T = 0$, and has rank at most $2N$ by construction. To see that its rank cannot be larger than $2\lfloor N_f/2 \rfloor$ either, it

helps to solve the $F$-term equations explicitly [28], as we now review.

Let $\mathcal{H}_r$ denote the subvariety of the Higgs branch where $M$ has rank at most $2r$:

$$\mathcal{H}_r \cong \{M = -M^T \in \mathbb{C}^{2N_f \times 2N_f} \mid M^2 = 0,\ \mathrm{rk}(M) \leq 2r\}\,. \tag{4.3}$$

Up to gauge and $O(2N_f)$ flavour rotations, the vevs of quark chiral superfields $Q$ on $\mathcal{H}_r$ can be written as

$$Q = \begin{pmatrix} 1 & i & 0 & 0 \\ 0 & 0 & 1 & i \end{pmatrix} \otimes \mathrm{diag}(q_1,\dots,q_r,0,\dots,0)\,, \tag{4.4}$$

where $r \leq \min(N,\lfloor N_f/2\rfloor)$, $q_a \in \mathbb{R}^+$ and it is understood that, if necessary, the resulting matrix is padded with rows and columns of zeros to make a $2N \times 2N_f$ matrix. The upper bound on $r$ arises from fitting as many $2 \times 4$ matrices along the diagonal in the $2N \times 2N_f$ matrix $Q$. A generic vev (4.4) breaks the gauge symmetry spontaneously to $USp(2(N-r))$ and the flavour symmetry to $SU(2)^r \times O(2(N_f - 2r))$. The meson matrix is

$$M = \begin{pmatrix} 0 & 0 & 1 & i \\ 0 & 0 & i & -1 \\ -1 & -i & 0 & 0 \\ -i & 1 & 0 & 0 \end{pmatrix} \otimes \mathrm{diag}(q_1^2,\dots,q_r^2,0,\dots,0)\,, \tag{4.5}$$

where again rows and columns of zeros are added to make a $2N_f \times 2N_f$ matrix. The first matrix in the tensor product (4.5) is self-dual, squares to zero and has rank 2, hence $M^2 = 0$ and $\mathrm{rk}(M) \leq 2r$. Since $r \leq \min(N,\lfloor N_f/2\rfloor)$, we obtain the upper bound on the rank of the meson matrix on the full Higgs branch quoted in (4.1), with the identification $\mathcal{H} = \mathcal{H}_{\min(N,\lfloor N_f/2\rfloor)}$. Mathematically, the orbit of (4.5) under the full $O(2N_f)$ flavour symmetry defines the closure of the nilpotent orbit of $O(2N_f)$ associated to the $D$-partition $(2^r, 1^{N_f - 2r})$:

$$\mathcal{H}_r \cong \overline{\mathcal{O}}^{O(2N_f)}_{(2^r, 1^{N_f - 2r})} = \bigcup_{\ell=0}^{r} \mathcal{O}^{O(2N_f)}_{(2^\ell, 1^{N_f - 2\ell})}\,. \tag{4.6}$$

The last equality expresses the closure of the nilpotent orbit as the union of the orbit itself and all its suborbits (the nilpotent orbit $\mathcal{O}^{O(2N_f)}_{(2^\ell, 1^{N_f - 2\ell})}$ is defined as in (4.3) except that $\mathrm{rk}(M) = 2\ell$). It has quaternionic dimension $2rN_f - r(2r+1)$, as can be easily computed using the Higgs mechanism, and is isomorphic to the full Higgs branch of the $USp(2r)$ SQCD theory with $N_f$ flavours.

So far we have described the Higgs branch using the full flavour symmetry group $O(2N_f) = SO(2N_f) \rtimes Z_2$, but it is important to discuss the action of the two subgroups separately, since they behave differently. Indeed, while the $SO(2N_f)$ normal subgroup only acts on the Higgs branch, the discrete $\mathbb{Z}_2$ symmetry generated by parity in flavour space acts not only on the Higgs branch but also on the Coulomb branch [1], due to fermionic zero modes which make all the monopole operators $U_n$ and $V_n$ odd under flavour parity. Let us then determine how the Higgs branches (4.3), (4.6), which are closures of nilpotent orbits of $O(2N_f)$, decompose into nilpotent orbits of $SO(2N_f)$, and how these are acted upon by the flavour parity.

If $r \leq N_f/2$, the representative (4.5) of the nilpotent orbit has at least two rows and columns of zero. As a result, any reflection in flavour space can be undone by a rotation by $\pi$ in a 2-plane. Hence the nilpotent orbit of $O(2N_f)$ is just a single nilpotent orbit of $SO(2N_f)$, which is mapped into itself by the $\mathbb{Z}_2$ symmetry:

$$\mathcal{H}_r \cong \overline{\mathcal{O}}^{O(2N_f)}_{(2^r, 1^{N_f - 2r})} = \overline{\mathcal{O}}^{SO(2N_f)}_{(2^r, 1^{N_f - 2r})} \qquad (r < N_f/2)\,. \tag{4.7}$$

If instead $r = N_f/2 \leq N$, the meson matrix (4.5) has no rows and columns of zeros, and the action of a reflection in flavour space is not equivalent to that of a proper rotation. Hence the nilpotent orbit of $O(2N_f)$ decomposes into two isomorphic nilpotent orbits of $SO(2N_f)$, which are mapped into each other by the $\mathbb{Z}_2$ symmetry:[17]

$$\mathcal{H}_{N_f/2} \cong \overline{\mathcal{O}}^{O(2N_f)}_{(2^{N_f})} = \overline{\mathcal{O}}^{SO(2N_f)}_{(2^{N_f}),+} \cup \overline{\mathcal{O}}^{SO(2N_f)}_{(2^{N_f}),-} \equiv \mathcal{H}_{N_f/2,+} \cup \mathcal{H}_{N_f/2,-} \, . \tag{4.8}$$

This means that the Higgs branch of bad $USp(2N)$ SQCD theories with an even number of flavours $N_f \leq 2N$ splits into two isomorphic components $\mathcal{H}_{N_f/2,\pm}$. Classically these two components intersect at a common subvariety, $\mathcal{H}_{N_f/2-1}$. Quantum-mechanically, however, the story is different, as we now explain.

## 4.2 Mixed branches and full moduli space

The classically exact Higgs branches must intersect the quantum corrected Coulomb branches at its singular loci, since there are new light degrees of freedom in the form of massless hypermultiplets. To understand how Higgs and Coulomb branches intersect, we can combine the analysis of the Higgs branch that we have just reviewed with the singularity structure of the Coulomb branch that we have obtained in Section 3. There is an extra constraint, given by the unbroken gauge and flavour symmetries on the Higgs branch. At a generic point of the subvariety $\mathcal{H}_r$ of the Higgs branch where the meson has rank at most $2r$, the unbroken gauge symmetry is $USp(2(N-r))$, and there are $N_f - 2r$ massless fundamental hypermultiplets (in addition to the free hypermultiplets that parametrize the local geometry of this Higgs branch). Therefore $\mathcal{H}_r$ should be part of a mixed Higgs-Coulomb branch, whose $(N-r)$ quaternionic dimensional Coulomb factor $\mathcal{C}_{N-r}$ is a singular locus of the Coulomb branch of $USp(2N)$ SQCD with $2N_f$ flavours which is isomorphic to the Coulomb branch of $USp(2(N-r))$ SQCD with $N_f - 2r$ flavours. This matches precisely the codimension $r$ singular locus (3.5), (3.11) in the Coulomb branch, leading to the identification $\mathcal{C}_{N-r} = \mathcal{C}^{(r)}_{\text{sing}}$.

The special case where $N_f \leq 2N$ is even and $r = N_f/2$ deserves further comment. We have seen in (4.8) that the Higgs branch splits into two separate components which are mapped into each other by the $\mathbb{Z}_2$ symmetry. While these two Higgs branch components intersect classically, there is no reason why they should do so in the quantum theory, and indeed they do not. The codimension $N_f/2$ singular locus of the Coulomb branch is the union of two disjoint components $\mathcal{C}^{(N_f/2)}_{\text{sing}\pm}$ (3.13) isomorphic to the Coulomb branch of the pure $USp(2N-N_f)$ SYM theory, which are mapped into each other by the $\mathbb{Z}_2$ flavour symmetry. They must then be identified with the roots of the two components of the Higgs branch (4.8).

To summarize, if $N_f > 2N$ or $N_f$ is odd, the full moduli space of vacua of $USp(2N)$ SCQD with $N_f$ flavours is

$$\mathcal{M} = \bigcup_{r=0}^{\min(N, \lfloor \frac{N_f}{2} \rfloor)} \left( \mathcal{C}_{N-r} \times \mathcal{H}_r \right), \tag{4.9}$$

where the Coulomb factor $\mathcal{C}_{N-r} = \mathcal{C}^{(r)}_{\text{sing}}$ is given by (3.5), (3.11) and is isomorphic to the full Coulomb branch of $USp(2(N-r))$ with $N_f - 2r$ flavours, while the Higgs factor $\mathcal{H}_r$ is given by (4.3) and is isomorphic to the full Higgs branch of $USp(2r)$ with $N_f$ flavours. If instead

---

[17]The first component is the orbit under $SO(2N_f)$ of (4.5) with $r = N_f/2$. The second component is the orbit under $SO(2N_f)$ of (4.5) with $r = N_f/2$ and with one $4 \times 4$ block anti-self-dual rather than self-dual. Mathematically, a nilpotent orbit of $O(2N_f)$ splits into two isomorphic nilpotent orbits of $SO(2N_f)$ whenever the associated partition of $2N_f$ is *very even*, *i.e.* when each even part appears an even number of times [29, 30].

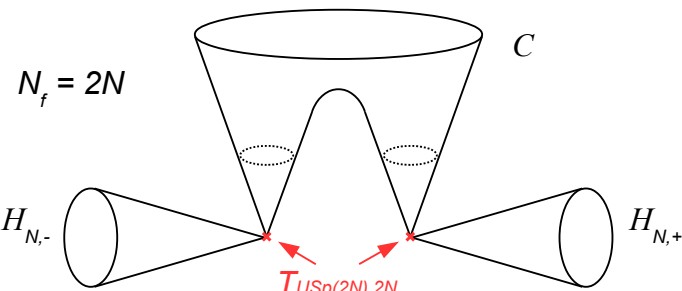

Figure 2: Schematic picture of the Coulomb and Higgs branches, $\mathcal{C}$ and $\mathcal{H}_{N,-} \cup \mathcal{H}_{N,+}$, of the $USp(2N)$ theory with $2N$ flavours. The two Higgs branch components meet the Coulomb branch at the two special singular points $\mathcal{C}^{*\pm}$ where the theory flows to $T_{USp(2N),2N}$.

$N_f \leq 2N$ and $N_f$ is even, the full moduli space of vacua is

$$\mathcal{M} = \left( \bigcup_{r=0}^{N_f/2-1} (\mathcal{C}_{N-r} \times \mathcal{H}_r) \right) \cup \left( \mathcal{C}^{(N_f/2)}_{\text{sing}+} \times \mathcal{H}_{N_f/2,+} \right) \cup \left( \mathcal{C}^{(N_f/2)}_{\text{sing}-} \times \mathcal{H}_{N_f/2,-} \right) \tag{4.10}$$

where now the Coulomb factors $\mathcal{C}^{(N_f/2)}_{\text{sing}\pm}$ are given by (3.13) and are each isomorphic to the Coulomb branch of the $USp(2N-N_f)$ pure SYM theory, while the Higgs factors $\mathcal{H}_{N_f/2,\pm}$ are given by (4.8) and are the closures of the two nilpotent orbits of $SO(2N_f)$ associated to the very even partition $(2^{N_f})$.

For $N_f = 2N$ we observe that each of the two Higgs branch components $\mathcal{H}_{N_f/2,\pm}$ intersects the Coulomb branch at a different special singular point $\mathcal{C}^{*\pm} = \mathcal{C}^{(N)}_{\text{sing}\pm}$, where the theory flows to $T_{USp(2N),2N}$, therefore the Higgs branch of the $T_{USp(2N),2N}$ SCFT is the closure of a single $SO(4N)$ nilpotent orbit $\overline{\mathcal{O}}^{SO(4N)}_{(2^{2N})}$. This is illustrated in Figure 2.

## 5  An unusual realization of mirror symmetry

Mirror symmetry of three-dimensional $\mathcal{N} = 4$ theories is the statement that pairs of different UV gauge theories flow to the same IR fixed point, with the role of the $SU(2)_H$ and $SU(2)_C$ R-symmetries exchanged [15]. In particular the infrared Higgs branch and Coulomb branch of mirror dual theories are exchanged under the duality map. To be more precise, in all examples that we are aware of, the duality relates the fixed points sitting at the origin of the moduli space of *good* theories. In those cases the Coulomb branch is a hyperkähler cone and is algebraically identical to the Coulomb branch of the CFT at its origin.[18] On the other side the Higgs branch is also a hyperkähler cone and matches the CFT Higgs branch. Mirror symmetry exchanges the CFT Coulomb branches and Higgs branches. This translates into the exchange of the algebraic Higgs branch and Coulomb branch in the mirror-dual UV good theories.

The situation is less clear for bad theories. Here the Coulomb branch is not a cone and has no good notion of "origin". More importantly it is not algebraically isomorphic to any CFT Coulomb branches (which are cones). In a previous work [14], we found that the fixed points of bad $U(N)$ SQCD theories, namely the CFTs that one can reach at different locations

---

[18]The metric on the Coulomb branch however depends on the gauge couplings and the CFT metric is obtained by sending $g^2 \to \infty$.

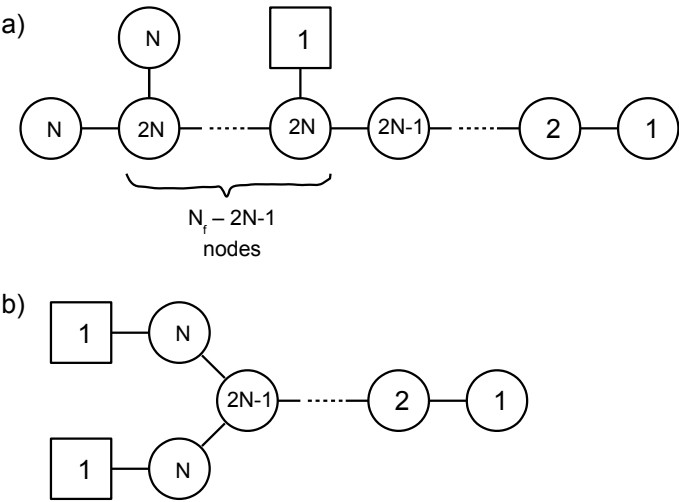

Figure 3: Quiver description of the mirror of $USp(2N)$ good theories with: a) $N_f \geq 2N + 2$ flavours; b) $N_f = 2N + 1$ flavours. Circles denote $U(n)$ gauge nodes and squares denote $U(m)$ flavour nodes.

in the Coulomb branch, all correspond to fixed points of good SQCD theories. This means that the local geometry around a CFT fixed point always reproduces the Coulomb branch of a good theory. Thus there is no new CFT genuinely associated to bad $U(N)$ SQCD theories, and therefore no new statement about mirror symmetry.

One could naively expect that these observations extend to all bad theories, in the sense that the fixed points that one can reach on their moduli space are always also fixed points of some good UV gauge theories. In the previous sections we have shown explicitly that this naive expectation is not quite correct. Instead we found a new set of fixed points $T_{USp(2N),2N}$ associated to the special singular loci of the bad $USp(2N)$ theories with $2N$ flavours. Moreover it was observed in [2] based on brane considerations that some special good quiver theories have naively a bad mirror dual, raising the question of how mirror symmetry is realized in those cases. In this section we find a mirror dual for the $T_{USp(2N),2N}$ fixed points that is a good quiver theory, realizing concretely a rather peculiar mirror symmetry scenario.

Let us first review mirror symmetry for the good theories, as found in [25, 31] using type IIB brane realizations. The mirror dual to the $USp(2N)$ theory with $N_f \geq 2N + 1$ flavours is a balanced flavoured $D$-shaped quiver theory with unitary gauge nodes. The cases $N_f = 2N + 1$ must be distinguished from the other cases $N_f \geq 2N + 2$. The mirror theories corresponding to these two classes are presented in Figure 3, using standard conventions for gauge nodes (circles) and flavour nodes (squares). Mirror symmetry predicts that the Higgs branch (respectively Coulomb branch) of these $D$-shaped quiver theories is isomorphic to the Coulomb branch (respectively Higgs branch) of the mirror $USp(2N)$ SQCD theories with $N_f$ flavours (see [32] for tests of this duality).

A mirror theory for the bad $USp(2N)$ theory with $2N$ flavours was also proposed in [25] as the balanced flavoured $D$-shaped quiver presented in Figure 4, which is a good theory. Our analysis shows that this cannot be a mirror dual in the same sense as for pairs of good theories, since the Coulomb branch of the $USp(2N)$ theory with $2N$ flavours is not isomorphic to the Higgs branch of the proposed mirror (which is a cone). Moreover the Higgs branch of the $USp(2N)$ theory splits into two separate components. This does not happens for the Coulomb branch of the $D$-shaped mirror. More simply, the bad $USp(2N)$ theory has two special points

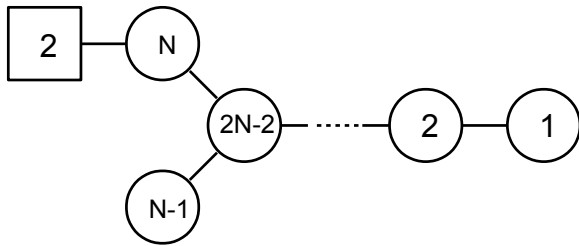

Figure 4: Mirror theory of the $T_{USp(2N),2N}$ CFT.

on its Coulomb branch, where the theory flows to the $T_{USp(2N),2N}$ CFT, while the $D$-shaped theory has only one origin on its moduli space.

Instead we propose the following scenario. Mirror symmetry is realized locally at each of the two special singular points $C^{*\pm}$. The CFT at the origin of the moduli space of the $D$-shaped theory is isomorphic to $T_{USp(2N),2N}$ upon exchanging the two $SU(2)$ R-symmetries. The Coulomb branch of the $D$-shaped theory is isomorphic to the Higgs component $\mathcal{H}_{N,+}$ (or $\mathcal{H}_{N,-}$) emanating from a special singular point. The Higgs branch of the $D$-shaped theory is isomorphic to the local Coulomb branch geometry near a special singular point (3.19). One may rephrase this by saying that the $D$-shaped quiver is the mirror of the $T_{USp(2N),2N}$ theory, where we imply that the Coulomb branch of $T_{USp(2N),2N}$ is described by the geometry (3.19) and its Higgs branch is $\mathcal{H}_{N,+}$.

As a first important check of our proposal we observe that the $T_{USp(2N),2N}$ Coulomb branch (3.19) has an $SU(2)_J$ global symmetry which was not present in the parent $USp(2N)$ theory. This maps in the mirror $D$-shaped quiver theory to the $SU(2)$ flavour symmetry which rotates the two $U(N)$ fundamental hypermultiplets.

As a further check we match the Hilbert series of the Coulomb branch of $T_{USp(2N),2N}$, which can be immediately extracted from its algebraic description presented in section 3.4, and the Hilbert series of the Higgs branch of the $D$-shaped theory. The details of the computation are given in appendix D. The equality between the Hilbert series follows from the very non-trivial identity (D.10). We perform the residue computation and arrive at a simpler form of the required identity, that we were able to check with Mathematica for values of $N$ up to 16.

## 6 Symmetric vacuum and sphere partition function

### 6.1 Symmetric vacuum

Recall that the $USp(2N)$ theory with $N_f$ flavours has an $O(2N_f)$ flavour symmetry acting on the $2N_f$ half-hypermultiplets. As a flavour symmetry, it naturally acts on the Higgs branch. However, as explained above, the $\mathbb{Z}_2$ parity inside $O(2N_f)$ also acts on the Coulomb branch, through its action on monopole operators (2.13) and masses,

$$
\begin{aligned}
\underline{\mathbb{Z}_2}: \quad & u_a^\pm \to -u_a^\pm, \\
& U_n, V_n \to -U_n, -V_n, \\
& \mathrm{Pf}(m) = \prod_\alpha m_\alpha \to -\mathrm{Pf}(m),
\end{aligned}
\tag{6.1}
$$

leaving the other generators invariant.

The global symmetries acting on the Coulomb branch of the massless theory are therefore $SU(2)_C \times \mathbb{Z}_2$, with only a subgroup $U(1)_R \times \mathbb{Z}_2$ visible in the algebraic description in a given complex structure. This global symmetry is completely broken at generic points on the Coulomb branch, since charged operators acquire vevs. For theories with $N_f > N$, there is a distinguished vacuum which preserves all the global symmetries. Following [14], we will call it the *symmetric vacuum $\mathcal{P}$*.

For good theories, $\mathcal{P}$ is the origin of the moduli space and the most singular point $\mathcal{C}^*$,

$$\underline{\text{Good theories } (N_f > 2N):} \quad \mathcal{P} = \{\widetilde{\Phi}_0 = 1, \text{others vanish}\}. \tag{6.2}$$

The infrared theory at $\mathcal{P}$ is the $T_{USp(2N),N_f}$ SCFT.

For bad theories, $\mathcal{P}$ is a generic point in the codimension $N_f - N - 1$ singular locus,

$$\underline{\text{Bad theories } (N < N_f \le 2N):} \quad \mathcal{P} = \{\widetilde{\Phi}_{2N-N_f} = (-1)^{N_f}, \text{others vanish}\}. \tag{6.3}$$

At low energies, the bad theory in the symmetric vacuum flows to the $T_{USp(2(N_f-N-1)),N_f}$ SCFT with $2N - N_f + 1$ decoupled free twisted hypermultiplets,

$$\mathcal{P} \xrightarrow{\text{IR}} T_{USp(2(N_f-N-1)),N_f} + (2N - N_f + 1) \text{ twisted hypers}. \tag{6.4}$$

Notice that $\mathcal{P}$ does not belong to the most singular locus and hence SCFTs of higher rank can be reached at more singular locations on the Coulomb branch.[19] The vacuum $\mathcal{P}$ is also described by the polynomials $U(w) = V(w) = 0$, $Q(w) = w^N$, $\widetilde{Q}(w) = w^{N_f-N-1}$, *both* for good and bad theories.

When $0 < N_f \le N$, there is no vacuum preserving the $U(1)_R \times \mathbb{Z}_2$ global symmetry. There is even no vacuum preserving the $\mathbb{Z}_2$ symmetry, which is therefore always spontaneously broken.

The relation between the bad $USp(2N)$ theory with $N_f > N$ flavours and the good $USp(2(N_f - N - 1))$ theory with $N_f$ flavours goes beyond the identification of their low-energy fixed point in the symmetric vacuum. The full Coulomb branch of the good theory $\mathcal{C}_g = \mathcal{C}_{USp(2(N_f-N-1)),N_f}$ is actually isomorphic to a subvariety of the Coulomb branch of the bad theory $\mathcal{C}_b = \mathcal{C}_{USp(2(N_f-N-1)),N_f}$,

$$\mathcal{C}_g \subset \mathcal{C}_b. \tag{6.5}$$

This is easily found from the polynomial relation (2.10) describing the CB of the bad theory. Restricting the degree of the polynomials $U(w), V(w)$ and $\widetilde{Q}(w)$ to be of degree $N_f - N - 2, N_f - N - 2$ and $N_f - N - 1$ respectively, the polynomial relation becomes that of the good theory, with the identification

$$\mathbf{U}(w) = \pm U(w), \quad \mathbf{V}(w) = \pm V(w), \quad \mathbf{Q}(w) = \widetilde{Q}(w), \quad \widetilde{\mathbf{Q}}(w) = Q(w),$$
$$w\mathbf{U}(w)^2 - \mathbf{V}(w)^2 - w^{N_f-1} + \mathbf{Q}(w)\widetilde{\mathbf{Q}}(w) = 0, \tag{6.6}$$

with the $\pm$ signs denoting several choices in the identification. Notice that with this identification $\widetilde{\mathbf{Q}}(w)$ is a monic polynomial. This is indeed always true for good theories.

The symmetric vacuum $\mathcal{P}$ is identified with the origin of $\mathcal{C}_g$. The operators which decouple at the symmetric vacuum and reduce to the $2N - N_f + 1$ free twisted hypermultiplets are the monopole operators $U_n$ and $V_n$ with $n = 0, 1, \cdots, 2N - N_f$, which do not participate in the map to the operators of the good theory.

From these results we can infer the relation between the IR and UV superconformal R-symmetry at the symmetric vacuum, in complete analogy with the analysis in [14]. In the IR theory at the symmetric vacuum the theory has R-symmetry $SU(2)_{IR}$ and an accidental global symmetry $SU(2N - N_f + 1)$ acting on the free hypermultiplets. The $SU(2)_{UV}$ R-symmetry is preserved in the symmetric vacuum and corresponds to the diagonal combination of $SU(2)_{IR}$ and the $SU(2)$ principal embedding inside $SU(2N - N_f + 1)$.

---

[19]For the extremal case $N_f = N + 1$ the symmetric vacuum $\mathcal{P}$ is a smooth point of the Coulomb branch and the infrared theory only contains free twisted hypermultiplets.

## 6.2 Sphere partition function

The previous results on the effective theory at the symmetric vacuum can be tested with exact supersymmetric localization computations. We consider theories in the range $N_f > N$ on the squashed three-sphere. The exact partition function of $\mathcal{N} = 2$ theories on squashed spheres was computed in [33] by supersymmetric localization. It is independent of the RG-flow and of the radius of the sphere (only the product of the radius with massive couplings appears in the partition function). At zero mass parameters, one might expect that the supersymmetric theory on the sphere has a single vacuum which preserves a $U(1)$ R-symmetry and the $O(2N_f)$ global symmetries of the theory. We will assume this in the following and check the consistency of this assumption *a posteriori*. This single vacuum must be continuously connected to the symmetric vacuum $\mathcal{P}$ of the flat space theory in the infinite radius limit.

We can therefore deform the $USp(2N)$ theory with $N < N_f \leq 2N$ in the symmetric vacuum to define it on the three-sphere. We have found that the low-energy theory in the symmetric vacuum can be described as the $T_{USp(2(N_f-N-1)),N_f}$ CFT, plus $2N - N_f + 1$ free twisted hypermultiplets, arising from $2(2N - N_f + 1)$ chiral operators of UV R-charge $r_n = N_f - 2N - 1 + n$, $n = 1, \cdots, 2(2N - N_f + 1)$. We therefore expect that the sphere partition function of the $USp(2N)$ theory with $N_f$ flavours matches the sphere partition function of the $USp(2(N_f - N - 1))$ theory with $N_f$ flavours, multiplied by the sphere partition function of the $2(2N - N_f + 1)$ free chiral multiplets with R-charges $r_n = N_f - 2N - 1 + n$,

$$Z_{USp(2N),N_f} = Z_{USp(2(N_f-N-1)),N_f} \prod_{n=1}^{2(2N-N_f+1)} Z_{\text{chiral}}(r_n). \tag{6.7}$$

This is our prediction and we will now check that this identity holds using localization results and known mathematical identities. In this formula we have used the UV R-charges, since the localization computation is done using UV R-charge assignments to the fields. Notice that, even though the above identity is suggestive of a Seiberg-like duality,[20] we have argued that it only follows from the expected low-energy behaviour of the bad $USp(2N)$ theory at the symmetric vacuum. The dual description involving the good $USp(2(N_f - N - 1))$ theory does not extend globally on the moduli space of vacua and therefore there is no duality. The Coulomb branches of the two theories are different and also the Higgs branches are different.

The partition function of the $USp(2N)$ theory with $N_f$ fundamental hypermultiplets on the squashed sphere takes the form [33, 36, 37]

$$Z = \frac{1}{\sqrt{-\omega_1 \omega_2}^N 2^N N!} \int \prod_{a=1}^{N} d\sigma_a \, Z_{\text{vec}}(\sigma) \prod_{\alpha=1}^{N_f} Z_{\text{hyper}}(\sigma, m_\alpha), \tag{6.8}$$

where $\omega_1, \omega_2$ are squashing parameters of the sphere,[21]

$$Z_{\text{vec}}(\sigma) = \prod_{a=1}^{N} \Gamma_h(\pm 2\sigma_a)^{-1} \prod_{a<b} \Gamma_h(\pm \sigma_a \pm \sigma_b)^{-1} \tag{6.9}$$

---

[20]An analogous Seiberg-like duality was proposed long ago by Aharony for 3d $\mathcal{N} = 2$ theories [34]. That duality was tested using sphere partition functions in [35] using the same mathematical identities which we will use below. It would be interesting to understand the $\mathcal{N} = 2$ duality from the low energy physics of the symmetric vacuum of the $\mathcal{N} = 4$ theory softly broken to $\mathcal{N} = 2$. We note here that the identity (6.13), which will be used in the following, suggests that the free fields of the $\mathcal{N} = 4$ theory become the singlet in the $\mathcal{N} = 2$ magnetic theory which is dual to the monopole operator of the electric theory.

[21]The sign $\pm$ in the expressions means that we take the product of the factor with the two signs: $f(x \pm y) = f(x + y)f(x - y)$.

is the contribution of the $\mathcal{N} = 4$ vector multiplet, and

$$Z_{\text{hyper}}(\sigma, m) = \prod_{a=1}^{N} \Gamma_h(\omega/2 \pm \sigma_a \pm m), \tag{6.10}$$

is the contribution of a fundamental hypermultiplet of real mass $m$. Here $\Gamma_h(x) \equiv \Gamma_h(x; \omega_1, \omega_2)$ denotes the hyperbolic gamma function and $\omega = (\omega_1 + \omega_2)/2$.

The contour integral for the matrix model of good theories is $\sigma \in \mathbb{R}^N$. For bad theories this naive contour integral is not convergent and one has to find a suitable contour. This has been achieved in the mathematics thesis [38] for bad theories with $N < N_f$. We will assume that the chosen contour, which makes the integral convergent and analytic in the various parameters, provides the correct physical evaluation of the sphere partition function of the bad theory.

The matrix model computing the sphere partition function (6.8) of the $USp(2N)$ theory with $N_f > N$ flavours defines the quantity $I_{n,2}^m(\mu)$ in [38] (see Def. 5.3.1, 5.3.17 and 5.3.15), with $n = N$, $m = N_f - N - 1$ and $\mu_{2\alpha-1} = \omega/2 + m_\alpha$, $\mu_{2\alpha} = \omega/2 - m_\alpha$, for $\alpha = 1, \cdots, n + m + 1 (= N_f)$. We claim that Theorem 5.5.9 in [38] is the identity (6.7) that we wanted to prove. This theorem reads

$$I_{n,2}^m(\mu) = I_{m,2}^n(\omega - \mu) \Gamma_h(2(m+1)\omega - \sum_r \mu_r) \prod_{r<s} \Gamma_h(\mu_r + \mu_s), \tag{6.11}$$

with the range of $r, s$ being 1 to $2n + 2m + 2$. Under the identifications with the gauge theory parameters, we obtain

$$Z_{USp(2N), N_f}(m) = Z_{USp(2(N_f - N - 1)), N_f}(m) \Gamma_h(\omega(N_f - 2N)), \tag{6.12}$$

where we have used the identity $\Gamma_h(\omega + x)\Gamma_h(\omega - x) = 1$ and $\Gamma_h(\omega) = 1$. To reach the identity (6.7), we set the real masses $m_\alpha$ to zero and use the identity

$$\Gamma_h(\omega(N_f - 2N)) = \prod_{n=1}^{2(2N - N_f + 1)} \Gamma_h(\omega r_n) = \prod_{n=1}^{2(2N - N_f + 1)} Z_{\text{chiral}}(r_n). \tag{6.13}$$

Note that the identity holds at non-zero real mass parameters $m_\alpha$ as well. It is easy to include small complex masses (which combine with real masses to form triplets under the $SU(2)$ R-symmetry acting on the Coulomb branch) in the analysis of the Coulomb branch and find that the infrared duality relation in the vicinity of the symmetric vacuum confirms the map between the masses of the bad and good theories.

The identity (6.7) is very non-trivial and provides a strong confirmation of our results, especially about the infrared behaviour at the symmetric vacuum.

## Acknowledgements

We thank Mathew Bullimore, Santiago Cabrera, Simone Giacomelli, Amihay Hanany and Peter Koroteev for fruitful discussions at various stages of the project.

## A  From $U(2)$ to $SU(2)$ with $N_f$ massive flavours

In this appendix we construct the CB relation for $SU(2)$ with $N_f$ massive flavours, starting from $U(2)$ with the same number of flavours.[22] The purpose of this exercise is to show that

---

[22]This approach was also used in [39] to analyse the Coulomb branch of $SU(2)$ SCQD theories with two and four flavours.

the resulting CB relation is a deformation of the CB equation of the massless theory, which is obtained by adding lower order terms, and to obtain the defining equation of the Atiyah-Hitchin manifold for $N_f = 0$. The final results provide confirmation for the derivation of the CB relations in Section 2, in the case of rank one.

We start from the generating polynomial of CB relations for $U(2)$ with $N_f$ flavours:

$$U^+(z)U^-(z) - P(z) = Q(z)\widetilde{Q}(z) \,, \tag{A.1}$$

where

$$
\begin{aligned}
U^\pm(z) &= V_0^\pm z - V_1^\pm \,, & Q(z) &= z^2 - \Phi_1 z + \Phi_2 \,, \\
\widetilde{Q}(z) &= \sum_{n=0}^{\widetilde{N}} (-1)^n \widetilde{\Phi}_n z^{\widetilde{N}-n} \,, & P(z) &= \sum_{n=0}^{N_f} (-1)^n M_n z^{N_f-n}
\end{aligned}
\tag{A.2}
$$

with $M_0 = 1$ and $\widetilde{Q}(z)$ is a polynomial of degree $\widetilde{N} = \max(N_f - 2, 0)$. Requiring that (A.1) holds for all values of $z$ determines the coefficients $\widetilde{\Phi}_n$ of $\widetilde{Q}(z)$ and imposes two relations on the remaining generators $\Phi_1$, $\Phi_2$, $V_0^\pm$ and $V_1^\pm$.

In order to go from $U(2)$ to $SU(2)$ gauge group, we gauge the $U(1)_J$ topological symmetry under which monopole operators have charges $J[V_n^\pm] = \pm 1$. The $U(1)_J$ invariants are simply $v_{ij} = V_i^+ V_j^-$ $(i, j = 0, 1)$, which satisfy the relation $\det(v) = v_{00}v_{11} - v_{01}v_{10} = 0$. In addition, the complex moment map equation in the corresponding hyperkähler quotient sets $\Phi_1 = 0$.[23] The CB relations for $U(2)$ that result from (A.1) are linear in $v_{ij}$ and determine $s \equiv v_{01} + v_{10}$ and $v_{11}$ in terms of $W \equiv -\Phi_2$ and $u = v_{00}$ as follows:

$$
\begin{aligned}
s &= -P_{\mathrm{odd}}(W) \equiv -P^-(z)/z|_{z^2=W} \,, \\
v_{11} &= -uW + P_{\mathrm{even}}(W) \equiv -uW + P^+(z)|_{z^2=W} \,,
\end{aligned}
\tag{A.3}
$$

where $P^\pm(z) = (P(z) \pm P(-z))/2$ are the even and odd parts of $P(z)$ under $z \to -z$.

The remaining generator $d \equiv v_{01} - v_{10}$ is not determined by the CB relations of $U(2)$. Using (A.3), the relation (A.3) can be rewritten in terms of $d$, $u$ and $W$ as

$$d^2 = P_{\mathrm{odd}}(W)^2 + 4u(uW - P_{\mathrm{even}}(W)) \,. \tag{A.4}$$

Next we change variable from $u$ to

$$U = 2u - \frac{P_{\mathrm{even}}(W) - P_{\mathrm{even}}(0)}{W} \,. \tag{A.5}$$

Note that the shift in the right-hand-side is by a polynomial in $W$, since the numerator of the second term is proportional to $W$. Substituting in the relation (A.4), we obtain

$$d^2 = U^2 W - 2P_{\mathrm{even}}(0)U - \left[\frac{1}{W}(P_{\mathrm{even}}(W)^2 - P_{\mathrm{even}}(0)^2) - P_{\mathrm{odd}}(W)^2\right] \,, \tag{A.6}$$

where $P_{\mathrm{even}}(0) = P^+(0) = P(0) = (-1)^{N_f} M_{N_f} = (-1)^{N_f} \prod_{\alpha=1}^{N_f} m_\alpha$. Upon expressing $P_{\mathrm{even}}(W)$ and $P_{\mathrm{odd}}(W)$ in terms of $P^\pm(z)$ (with $W = z^2$), the terms inside the square brackets in (A.6) are easily seen to be equal to

$$(-1)^{N_f} \frac{\widetilde{P}(W) - \widetilde{P}(0)}{W} = (-1)^{N_f} \sum_{n=0}^{N_f-1} (-1)^n \widetilde{M}_n W^{N_f-1-n} \,, \tag{A.7}$$

---

[23] Shifting the level of the moment map to $\Phi_1 = \mu$ can be undone by an appropriate redefinition.

where $\widetilde{P}(W) = \prod_{\alpha=1}^{N_f}(W - m_\alpha^2) \equiv \sum_{n=0}^{N_f}(-1)^n \widetilde{M}_n W^{N_f-n}$ is the characteristic polynomial of the $O(2N_f)$ flavour symmetry of $SU(2)$ with $2N_f$ doublet half-hypermultiplets. We reach therefore the final form of the Coulomb branch relation for $SU(2)$ with $N_f$ flavours, which agrees with (2.8) for $N = 1$ and $w = W = \varphi^2$:

$$d^2 = U^2 W - 2(-1)^{N_f}\prod_{\alpha=1}^{N_f} m_\alpha \cdot U - (-1)^{N_f}\frac{\widetilde{P}(W) - \widetilde{P}(0)}{W}. \tag{A.8}$$

In the massless limit, the Coulomb branch relation reduces to

$$\begin{aligned}
N_f = 0: \qquad & d^2 = U^2 W - 2U \\
N_f > 0: \qquad & d^2 = U^2 W - (-1)^{N_f} W^{N_f-1}.
\end{aligned} \tag{A.9}$$

# B  Geometry near special singular points

The CB relations for the $USp(2N)$ theory with $N_f = 2N$ flavours are

$$\sum_{n_1+n_2=k}(U_{n_1}U_{n_2} + \Phi_{n_1}\widetilde{\Phi}_{n_2}) + \sum_{n_1+n_2=k-1}V_{n_1}V_{n_2} = \delta_{k,0}, \tag{B.1}$$

for $k = 0, \cdots, 2N - 1$. The special singular loci are the two isolated points $\mathcal{C}^{*\pm} = \{(U_n, V_n, \Phi_{n+1}, \widetilde{\Phi}_n) = (\pm\delta_{n,0}, 0, 0, 0)|n = 0, \cdots, N - 1\}$. To obtain the approximate CB relations near $\mathcal{C}^{*+}$, we can first solve for $U_0$ in a neighborhood of $\widetilde{\Phi}_0 = 0$ using the $k = 0$ relation,

$$U_0 = \sqrt{1 - \widetilde{\Phi}_0}. \tag{B.2}$$

Note that $\sqrt{1 - \widetilde{\Phi}_0} = 1 - \frac{1}{2}\widetilde{\Phi}_0 + O(\widetilde{\Phi}_0^2)$ is a holomorphic single-valued function of $\widetilde{\Phi}_0$ in a neighborhood of $\widetilde{\Phi}_0 = 0$ (there is no branch cut). The remaining relations become

$$\begin{aligned}
&\underline{k = 1, \cdots, N - 1:} \\
&2U_k\sqrt{1 - \widetilde{\Phi}_0} + \widetilde{\Phi}_k + \Phi_k\widetilde{\Phi}_0 + \sum_{\substack{n_1+n_2=k \\ n_1,n_2 \geq 1}}(U_{n_1}U_{n_2} + \Phi_{n_1}\widetilde{\Phi}_{n_2}) + \sum_{n_1+n_2=k-1}V_{n_1}V_{n_2} = 0, \\
&\underline{k = N, \cdots, 2N - 1:} \sum_{n_1+n_2=k}(U_{n_1}U_{n_2} + \Phi_{n_1}\widetilde{\Phi}_{n_2}) + \sum_{n_1+n_2=k-1}V_{n_1}V_{n_2} = 0.
\end{aligned} \tag{B.3}$$

It is not obvious which terms can be dropped in the limit of small operator vevs, however we know that the limiting geometry should have a $U(1)_{\mathbb{C}}^{\mathrm{IR}} \cong \mathbb{C}^*$ action with all generators having charges bigger or equal to one.[24] This $U(1)_{\mathbb{C}}^{\mathrm{IR}}$ cannot corresponds to the $U(1)^{\mathrm{UV}}$ acting on the full Coulomb branch, since $\widetilde{\Phi}_0$ has charge zero under $U(1)_{\mathbb{C}}^{\mathrm{UV}}$. Therefore there should be a change of variables in the vicinity of the special singular locus, which makes apparent an emergent $U(1)_{\mathbb{C}}^{\mathrm{IR}}$ symmetry as we zoom in on the special vacuum. Concretely, we are looking for a change of variables in which a $U(1)$ global symmetry is manifest in the limit of small operators vevs. We propose the change of variables

$$\begin{aligned}
U_n' &= U_n + f(\widetilde{\Phi}_0)\widetilde{\Phi}_n, \\
\Phi_n' &= \Phi_n - 2f(\widetilde{\Phi}_0)U_n - f(\widetilde{\Phi}_0)^2\widetilde{\Phi}_n,
\end{aligned} \tag{B.4}$$

---

[24]$U(1)_{\mathbb{C}}$ is the complexification of $U(1)_R \subset SU(2)_C$ by the dilatation symmetry of the SCFT.

for $n = 1, \cdots, N-1$, where $f$ is to be fixed so that a $U(1)$ symmetry emerges, and we use the convention that $\Phi'_N = \Phi_N$. The advantage of this change of variables is that it satisfies

$$U_{n_1} U_{n_2} + \frac{1}{2}(\Phi_{n_1}\widetilde{\Phi}_{n_2} + \Phi_{n_2}\widetilde{\Phi}_{n_1}) = U'_{n_1} U'_{n_2} + \frac{1}{2}(\Phi'_{n_1}\widetilde{\Phi}_{n_2} + \Phi'_{n_2}\widetilde{\Phi}_{n_1}), \tag{B.5}$$

for $1 \le n_1, n_2 \le N-1$, leaving some terms in the CB relations invariant . After the change of variables we obtain

$$
\begin{aligned}
\underline{k = 1, \cdots, N-1}: \quad & 2U'_k\big(\sqrt{1-\widetilde{\Phi}_0} + f(\widetilde{\Phi}_0)\widetilde{\Phi}_0\big) + \Phi'_k\widetilde{\Phi}_0 + \widetilde{\Phi}_k\Theta \\
& + \sum_{\substack{n_1+n_2=k \\ n_1,n_2\ge 1}} (U'_{n_1} U'_{n_2} + \Phi'_{n_1}\widetilde{\Phi}_{n_2}) + \sum_{n_1+n_2=k-1} V_{n_1} V_{n_2} = 0, \\
\underline{k = N, \cdots, 2N-1}: \quad & \sum_{n_1+n_2=k} (U'_{n_1} U'_{n_2} + \Phi'_{n_1}\widetilde{\Phi}_{n_2}) + \sum_{n_1+n_2=k-1} V_{n_1} V_{n_2} = 0,
\end{aligned}
\tag{B.6}
$$

with $\Theta = 1 + 2(\widetilde{\Phi}_0 - \sqrt{1-\widetilde{\Phi}_0})f(\widetilde{\Phi}_0) + \widetilde{\Phi}_0 f(\widetilde{\Phi}_0)^2$. The above equations would have an emergent $U(1)$ symmetry acting on $\Phi'_n$ and $\widetilde{\Phi}_n$ with opposite charges in the limit of small $\widetilde{\Phi}_0$, if not for the $\widetilde{\Phi}_k\Theta$ term. We thus choose $f$, holomorphic in a neighborhood of zero, such that $\Theta = 0$. We are interested in the local geometry near the special singular point, which has $\widetilde{\Phi}_0 = 0$, thus we can expand the expression to linear order in $\widetilde{\Phi}_0$. In this approximation $f(\widetilde{\Phi}) = \frac{1}{2} + \frac{7}{8}\widetilde{\Phi}_0 + O(\widetilde{\Phi}_0^2)$, and we observe that the factor $\big(\sqrt{1-\widetilde{\Phi}_0} + f(\widetilde{\Phi}_0)\widetilde{\Phi}_0\big)$ appearing in the relations nicely simplifies to $1 + O(\widetilde{\Phi}_0^2)$. Therefore, neglecting $O(\widetilde{\Phi}_0^2)$ corrections, which are irrelevant in an infinitesimal neigbourhood of the special singular point, we obtain the final CB relations of the local geometry near the special singular point,

$$
\begin{aligned}
& \underline{k = 1, \cdots, N-1}: \\
& 2U'_k + \sum_{n_1+n_2=k} (U'_{n_1} U'_{n_2} + \Phi'_{n_1}\widetilde{\Phi}_{n_2}) + \sum_{n_1+n_2=k-1} V_{n_1} V_{n_2} = 0, \\
& \underline{k = N, \cdots, 2N-1}: \quad \sum_{n_1+n_2=k} (U'_{n_1} U'_{n_2} + \Phi'_{n_1}\widetilde{\Phi}_{n_2}) + \sum_{n_1+n_2=k-1} V_{n_1} V_{n_2} = 0,
\end{aligned}
\tag{B.7}
$$

where we reincorporated the term $\Phi'_k\widetilde{\Phi}_0$ in the sum over $n_1, n_2$, in the first line. These CB relations indeed have an accidental global symmetry acting on $\Phi'_n$ and $\widetilde{\Phi}_n$ generators with opposite charges, that can be used to assign positive $U(1)^{\mathrm{IR}}_{\mathbb{C}}$ charges to all generators.

Introducing $U'_0 = 1$ we can write the CB relations in a single set as

$$\underline{k = 1, \cdots, 2N-1}: \quad \sum_{n_1+n_2=k} (U'_{n_1} U'_{n_2} + \Phi'_{n_1}\widetilde{\Phi}_{n_2}) + \sum_{n_1+n_2=k-1} V_{n_1} V_{n_2} = 0. \tag{B.8}$$

To make the global symmetries even more manifest it will also be convenient to redefine $\Phi'_n \to \Phi'_{n-1}$, so that we now have $N$ $SU(2)_J$ triplets of generators $(\Phi'_n, V_n, \widetilde{\Phi}_n)^{N-1}_{n=0}$ of IR R-charge $2n+1$ for $n = 0, 1, \ldots, N-1$, and the CB relations are the $SU(2)_J$ singlets,

$$\underline{k = 1, \cdots, 2N-1}: \quad \sum_{n_1+n_2=k} U'_{n_1} U'_{n_2} + \sum_{n_1+n_2=k-1} (\Phi'_{n_1}\widetilde{\Phi}_{n_2} + V_{n_1} V_{n_2}) = 0. \tag{B.9}$$

## C  Coulomb branch relations without auxiliary generators

The method described in this paper to obtain the Coulomb branch of $USp(2N)$ SQCD theories leads to algebraic descriptions involving auxiliary generators ($\widetilde{\Phi}_n$ or $U'_n$). With these additional

generators the CB relations take a simple form, which is only quadratic in the generators. However it would also be useful to have a description with a minimal number of generators and therefore to solve for the auxiliary ones. In this appendix we show how to solve for the auxiliary generators, focusing on the Coulomb branch of the $T_{USp(2N),2N}$ theory for concreteness.

The CB relations are encoded into the generating polynomial relation (3.23), which involves the auxiliary generators $U'_n$ as the coefficients of the polynomial $U'(w)$. It can be solved for $U'(w)$, giving

$$U'(w) = w^{N-1}\left(1 + w^{1-2N}S(w)\right)^{1/2} = w^{N-1}\sum_{k=0}^{\infty}\binom{\frac{1}{2}}{k}\left(w^{1-2N}S(w)\right)^k , \tag{C.1}$$

where we expanded near $w = \infty$ in the last equality and

$$S(w) = Q'(w)\widetilde{Q}(w) + V^2(w) = \sum_{n=0}^{2N-2}(-1)^n S_n w^{2N-2-n} \tag{C.2}$$

is a polynomial of degree $2N - 2$ in $w$, with

$$S_n = \sum_{n_1+n_2=n}(\Phi'_{n_1}\widetilde{\Phi}_{n_2} + V_{n_1}V_{n_2}) . \tag{C.3}$$

Requiring that (C.1) be a polynomial (of degree $N-1$) in $w$ provides an alternative and explicit derivation of the CB relations: the polynomial part of (C.1) gives the actual polynomial $U'(w)$, while setting to zero all the negative Laurent coefficients of (C.1) imposes the ideal of CB relations among the generators $(\Phi'_n, V_n, \widetilde{\Phi}_n)_{n=0}^{N-1}$. It turns out that this ideal is generated by the first $N$ Laurent coefficients of (C.1), whereas the coefficients of $w^{-h}$ with $h > N$ are generated. Explicitly, if we define

$$S_n^{(k)} = \begin{cases} 0 & n < 0 \\ \sum_{n_1+\cdots+n_k=n} S_{n_1}\ldots S_{n_k} & n \geq 0 \end{cases} \tag{C.4}$$

for $k \geq 1$ and $S_n^{(0)} = \delta_{n,0}$, so that $S(w)^k = \sum_{n=0}^{2k(N-1)}(-1)^n S_n^{(k)}w^{2k(N-1)-n}$, we obtain the polynomial

$$U'(w) = \sum_{n=0}^{N-1}(-1)^n w^{N-1-n}\sum_{k=0}^{n}(-1)^k\binom{\frac{1}{2}}{k}S_{n-k}^{(k)} , \tag{C.5}$$

and the ideal of $N$ CB relations can be written compactly as

$$\sum_{k=1}^{N-1+h}(-1)^k\binom{\frac{1}{2}}{k}S_{N-1-k+h}^{(k)} = 0 , \qquad h = 1,\ldots,N . \tag{C.6}$$

Note that this presentation of the CB ideal is not identical to the one that is obtained by substituting the polynomial (C.5) in (3.23), but the two ideals coincide. (C.6) is not in Gröbner basis form, but defines the ideal in a very compact way. Conversely, substituting (C.5) in (3.23) appears to yield the ideal in Gröbner basis form, but does not have as simple a presentation as (C.6).

# D  Test of mirror symmetry for $T_{USp(2N),2N}$ with Hilbert series

The quiver

$$\begin{array}{c} \underset{N-1}{\circ} \\ | \\ \underset{2}{\square} - \underset{N}{\circ} - \underset{2N-2}{\circ} - \underset{2N-3}{\circ} - \cdots - \underset{2}{\circ} - \underset{1}{\circ} \end{array} \tag{D.1}$$

was proposed in [25] as a mirror of $USp(2N)$ SQCD with $2N$ fundamental flavours. More precisely, [25] pointed out that the Higgs branch of $USp(2N)$ SQCD with $2N$ flavours is the union of two isomorphic cones associated to the two spinor representations of $D_{2N}$, and the Coulomb branch of the "mirror" quiver (D.1) is isomorphic to either of the two cones. We have seen in Section 3.3 that the roots of the two top-dimensional Higgs cones on the Coulomb branch of $USp(2N)$ SQCD with $2N$ flavours are separated quantum-mechanically, and are mapped into one another by the $\mathbb{Z}_2$ parity inside the $O(4N)$ flavour symmetry, which changes sign to the monopole operators of smallest magnetic charge [1]. We thus interpreted the quiver (D.1) as the mirror of the IR SCFT $T_{USp(2N),2N}$ sitting at either of the roots of the two top-dimensional Higgs branch components. In this appendix we successfully compare the Higgs branch of the quiver (D.1) with the local Coulomb branch geometry (3.19) of $USp(2N)$ SQCD with $2N$ flavours near one of the two roots, showing that the Hilbert series of the two varieties coincide.

Let us first extract the Hilbert series of the local Coulomb branch geometry near one of the two roots of the Higgs branch, using the explicit local Coulomb branch relations (3.19) and the IR R-charges (3.20). The first $N-1$ Coulomb branch relations (3.19) allow us to solve for the generators $U'_n$, leaving an $SU(2)$ triplet of generators $(\Phi'_n, V_n, \widetilde{\Phi}_n)$ at IR R-charge $2n+1$ for all $n = 0, 1, \ldots, N-1$. The rest of (3.19) then describes $N$ algebraically independent $SU(2)$ singlet relations among these $3N$ generators, at IR R-charges $2k$ with $k = N, N+1 \ldots, 2N-1$. The Hilbert series of the local Coulomb branch geometry of $USp(2N)$ SQCD with $2N$ flavours near either of the two Higgs branch roots is therefore

$$H_{CB}(t; w) = \mathrm{PE}\left[[2]_w \sum_{n=0}^{N-1} t^{2n+1} - \sum_{k=N}^{2N-1} t^{2k}\right], \tag{D.2}$$

where $[2]_w = w^2 + 1 + w^{-2}$ is the character of the triplet representation of $SU(2)$, and PE denotes the plethystic exponential, which generates symmetric products.[25]

In order to compute the Hilbert series of the Higgs branch of (D.1), we will first use the Hilbert series of the Higgs branch of the $T[SU(2N-1)]$ theory, described by the quiver

$$\underset{2N-1}{\square} - \underset{2N-2}{\circ} - \underset{2N-3}{\circ} - \cdots - \underset{2}{\circ} - \underset{1}{\circ}. \tag{D.3}$$

The Hilbert series of the Higgs branch of $T[SU(2N-1)]$ is [7]

$$H[T(SU(2N-1))](t; x) = \mathrm{PE}\left[t \sum_{i,j=1}^{2N-1} x_i x_j^{-1} - \sum_{n=1}^{2N-1} t^n\right]. \tag{D.4}$$

This takes into account the Molien integral over the $2N-2$ rightmost gauge nodes.

We then decompose $U(2N-1) \supset U(N-1) \times U(N)$, and let $x_i = z_i$ for $i = 1, \ldots, N-1$ be fugacities for the $U(N-1)$ gauge node above the $U(2N-2)$ node in the quiver (D.1), and $x_{N+j} = y_j$ for $j = 1, \ldots, N$ be fugacities for the $U(N)$ gauge node to the left of the $U(2N-2)$ gauge node. Taking into account the contributions of $F$-term relations due to the complex adjoint scalars of $U(N-1) \times U(N)$, the Hilbert series of the Higgs branch of the quiver (D.1)

---

[25]The plethystic exponential (PE) of a multi-variate function $f(x_1, \ldots, x_n)$ is defined by

$$\mathrm{PE}[f(x_1, \ldots, x_n)] = \exp\left(\sum_{p=1}^{\infty} \frac{1}{p} f(x_1^p, \ldots, x_n^p)\right).$$

reads

$$H_{HB}(t;w) = \oint d\mu_y^{U(N)} \oint d\mu_z^{U(N-1)}$$
$$\mathrm{PE}\left[ t \sum_{i=1}^{N-1}\sum_{j=1}^{N}\left(\frac{z_i}{y_j} + \frac{y_j}{z_i}\right) + t^{1/2}\sum_{j=1}^{N}\sum_{a=1}^{2}\left(\frac{y_j}{w_a} + \frac{w_a}{y_j}\right) - \sum_{n=1}^{2N-1} t^n \right],$$

(D.5)

where $w_1 \equiv w$ and $w_2 \equiv 1/w$ are flavour fugacities and $\oint d\mu_y^{U(N)} \equiv \oint \frac{1}{N!}\prod_{i\neq j}(1 - y_i/y_j)$ denotes the Molien integral over $U(N)$ with fugacities $y$, with Haar measure. The Molien integral over $U(N-1)$ can be easily computed, yielding

$$H_{HB}(t;w) = \oint d\mu_y^{U(N)} \mathrm{PE}\left[ t^{1/2}\sum_{j=1}^{N}\sum_{a=1}^{2}\left(\frac{y_j}{w_a} + \frac{w_a}{y_j}\right) - \sum_{n=1}^{2N} t^n + t^2 \sum_{i,j=1}^{N}\frac{y_i}{y_j} \right].$$

(D.6)

(The Molien integral over the $z$ fugacities is the same as the Hilbert series the moduli space of $U(N-1)$ SQCD with $N$ flavours and four supercharges, which is generated by an $N \times N$ meson matrix with zero determinant.)

Upon factoring out $\mathrm{PE}[-\sum_{n=1}^{2N} t^n]$, the final integral over $U(N)$ with fugacities $y$ can also be interpreted as the Hilbert series of the moduli space of a theory with four supercharges, but now with gauge group $U(N)$, two fundamental flavours of R-charge 1/2 and an adjoint of R-charge 2. The refined Hilbert series of that theory takes the form

$$H_{U(N),adj,2}(\tau;s,w,\widetilde{w}) = \oint d\mu_y^{U(N)} \mathrm{PE}\left[ \tau \sum_{j=1}^{N}\sum_{a=1}^{2}\left(\frac{y_j}{\widetilde{w}_a} + \frac{w_a}{y_j}\right) + \tau s \sum_{i,j=1}^{N}\frac{y_i}{y_j} \right],$$

(D.7)

where $\tau$ is the R-symmetry fugacity, $s/\tau^3$ is the adjoint flavour fugacity and $w_a, \widetilde{w}_a$ are the (anti-)fundamental flavour fugacities. The explicit residue computation[26] yields contributions from the sets of poles

$$y_i = w_1 \tau(\tau s)^{i-1}, \quad i = 1, \cdots, p,$$
$$y_{p+i} = w_2 \tau(\tau s)^{i-1}, \quad i = 1, \cdots, N-p,$$

(D.8)

with $p = 0, 1, \cdots, N$, and counted $N!$ times from permutations of the $y_i$. The Hilbert series then evaluates to

$$H_{U(N),adj,2}(\tau;s,w,\widetilde{w})$$
$$= \sum_{p=0}^{N} \frac{1}{\prod_{i=1}^{p}\left(1-(\tau s)^i\right)\left(1-\frac{w_1}{\widetilde{w}_1}\frac{\tau}{s}(\tau s)^i\right)\left(1-\frac{w_1}{\widetilde{w}_2}\frac{\tau}{s}(\tau s)^i\right)\left(1-\frac{w_2}{w_1}(\tau s)^{N-p+1-i}\right)}$$
$$\times \frac{1}{\prod_{i=1}^{N-p}\left(1-(\tau s)^i\right)\left(1-\frac{w_2}{\widetilde{w}_1}\frac{\tau}{s}(\tau s)^i\right)\left(1-\frac{w_2}{\widetilde{w}_2}\frac{\tau}{s}(\tau s)^i\right)\left(1-\frac{w_1}{w_2}(\tau s)^{p+1-i}\right)}.$$

(D.9)

Explicit computations up to $N = 16$ using Mathematica give (very strong) support for the identity

$$H_{U(N),adj,2}(\tau;s,w,\widetilde{w}) = \mathrm{PE}\left[ \sum_{n=1}^{N}(\tau s)^n + \tau^2\left(\sum_{a,b=1}^{2}\frac{w_a}{\widetilde{w}_b}\right)\sum_{n=0}^{N-1}(\tau s)^n - \tau^4\frac{w_1 w_2}{\widetilde{w}_1 \widetilde{w}_2}\sum_{n=0}^{N-1}(\tau s)^{N-1+n} \right].$$

(D.10)

---

[26]The contours of integration are unit circles and the fugacities are taken to obey $|w_a| = |\widetilde{w}_a| = 1$, $|\tau| < 1$ and $|s| < 1$.

Using the result (D.9) with $\tau = t^{1/2}$, $s = t^{3/2}$ and $\widetilde{w}_a = w_a$, the Hilbert series (D.6) of the Higgs branch of the "mirror" quiver (D.1) can finally be written as

$$
\begin{aligned}
H_{HB}(t;w) &= \text{PE}\left[ -\sum_{n=1}^{2N} t^n + \sum_{n=1}^{N} t^{2n} + (1 + [2]_w)\sum_{n=0}^{N-1} t^{2n+1} - \sum_{n=0}^{N-1} t^{2(N+n)} \right] \\
&= \text{PE}\left[ [2]_w \sum_{n=0}^{N-1} t^{2n+1} - \sum_{k=N}^{2N-1} t^{2k} \right].
\end{aligned}
\tag{D.11}
$$

This precisely agrees with the Hilbert series (D.2) of the local geometry of the Coulomb branch of $USp(2N)$ SQCD with $2N$ flavours near the root of either Higgs cone.

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
