# Peer review of "The Infrared Fixed Points of 3d $\mathcal{N}=4$ $USp(2N)$ SQCD Theories"

_SciPost Physics, doi:SciPost Phys. 5, 015 (2018)_

## Round 2 · Referee Report · Peter Koroteev (Referee 1) · 2018-6-7

Strengths

  1. The paper discusses subtle aspects of nonperturbative physics of 3d $\mathcal{N}=4$ theories with symplectic gauge groups which has not been previously studied

  2. All nontrivial claims are supported by correct and explicit calculations

Weaknesses

  1. Due to the complexity some computations are hard to follow (see requested changes).

Report

The authors conduct an extensive study of infrared dynamics of 3d $\mathcal{N}=4$ SQCD with symplectic gauge groups. This manuscript is a follow-up work to the authors' previous paper on SQCDs with unitary groups. The results are solid, well presented and open to door for further work on (quiver) gauge theories with other gauge groups. The manuscript should be published once the requested changes below are made.

Requested changes

\begin{enumerate}
\item It is worth explaining a bit more how formula (2.1) appears from the $Sp(N)$ root lattice, etc.
\item The discussion starting from the middle of p.9 is about the continuation to the discriminant locus is hard to follow at times. Is it technically hard to do the analysis without dividing by common factors? The early reference to Appendix A makes it confusing and does not show why these branches are spurious. The strategy written on pp. 9-10 should be stated more clearly. (not important, but word `massage' is overused).

\item Section 5 about mirror symmetry.
It is not quite clear from the description how the mirror symmetry chooses between two components $\mathcal{H}_\pm$.

Also there are some papers by Hanany et al where a given D-shaped quiver with unitary groups has two mirrors -- good and bad which share the same Hilbert series. Using brane language one arrives to a bad mirror. Yet, there's a way to land on a good mirror. How does this observation connects with the scenario which the authors suggest in the middle of p.29?

\item Appendix B. Where does formula $f(\tilde\Phi)=\frac{1}{2}+\frac{7}{8}\tilde\Phi_0$ in the middle of the page 37 come from?

\end{enumerate}

  • validity: high
  • significance: high
  • originality: top
  • clarity: good
  • formatting: good
  • grammar: good

Author:  Benjamin Assel  on 2018-06-11  [id 272]

(in reply to Report 1 by Peter Koroteev on 2018-06-07)
Category:
answer to question

We thank the referee for his careful reading of the manuscript and for his comments. Below we provide answers for each issue raised in order.

  • We will add more details to explain how Equation (2.1) arises in a revised version.

  • It is simpler to do the analysis without dividing by common factors, but it leads to spurious branches of vacua. For instance a branch with all CB operators vanishing except for $U_0$ which is unconstrained. In this case the branch can be ruled out because it is not hyperk\"ahler (it has complex dimension one). We rule out such branches for $USp(2) = SU(2)$ SQCD theories for which an alternative construction is available (given in Appendix A indeed) and which does not have such extra branches. The prescription for higher rank theories is a conjecture based on a generalization of the $SU(2)$ case. We can try to improve on the current discussion by adding these more precise comments, but we feel that further comments might make the discussion heavier and harder to follow. We can add two more arguments in favour of our prescription, which we decided not to include in the paper for similar reasons. First of all, not dividing by common factors leads to chiral ring relations of higher R-charge, in conflict with Hilbert series results. Secondly, it is not hard to show that the Coulomb branch relations (2.8) of $USp(2N)$ SQCD, which we obtained after dividing by common factors in the abelianized relations, reduce to the well-known Coulomb branch relations of $U(N)$ SQCD in the limit of large and almost equal scalar vevs, as expected.

  • It is not clear to us which second mirror dual to the D-shaped quiver the referee refers to. There is a second class of good theories mirror dual to $USp(2N)$ SQCD fixed points, in the form of orthosymplectic quivers, which did not study in the paper. To answer the question, the two Higgs branches are isomorphic, as are the two SCFTs at the special singular points, therefore there is not really any meaning to saying which branch emanates from which singular point, or how to choose between branches. As an extra comment we can say that mirror symmetry is a statement of infrared duality in a given vacuum. One needs to choose a vacuum where the theory flows to an SCFT in the IR, and then ask what the mirror is. Our main point is that $USp(2N)$ SQCD with $2N$ flavors at any of the two special singular points has the same mirror, which we provide in the form of a D-shaped quiver. As we mention towards the mirror of page 6, there is an alternative mirror theory, in the form of a good ortho-symplectic linear quiver. It is natural to conjecture that this is also mirror to $USp(2N)$ SQCD with $2N$ flavors at any of the two special singular points.

  • The expansion $f(\tilde\Phi_0)=\frac{1}{2}+\frac{7}{8}\tilde\Phi_0 + O(\tilde\Phi_0)$ arises from solving the algebraic equation $\Theta=0$ order by order at small $\tilde\Phi_0$. We will make this clearer in the revised version.

Author:  Benjamin Assel  on 2018-06-25  [id 280]

(in reply to Benjamin Assel on 2018-06-11 [id 272])

We thank the referee for his reply. We will add the aforementioned comments to the appendix and in a footnote in the main text.

Peter Koroteev  on 2018-06-14  [id 273]

(in reply to Benjamin Assel on 2018-06-11 [id 272])
Category:
remark

I thank the authors for their reply.

Regrading the spurious factors, perhaps it would be sufficient to add the comments that the authors were referring to in the appendix. This way the readability of the main text won't be affected and the interested reader could learn more from the appendix.

The rest seems to be okay. The manuscript can be published after the above-mentioned corrections in the author's reply will be implemented.

---

## Round 2 · Referee Report · Anonymous (Referee 2) · 2018-6-20

Strengths

1- clear, well written 2- comprehensive

Weaknesses

I don't think it has any particular weaknesses. It is a technical subject, but I think the authors do a good job explaining what they did.

Report

This is an interesting paper describing the fixed points and moduli space of vacua of 3d ${\cal N} = 4$ gauge theories with $USp(2N)$ gauge group and $N_f$ fundamental hypermultiplet matter flavors.

The discussion on the Coulomb branch uses the technique developed by Bullimore-Dimofte-Gaiotto (ref. [12]) to determine the chiral ring relations. There is, however, a subtlety related how to continue the relations derived at a generic point on the Coulomb branch to points where there is non-Abelian symmetry enhancement. The authors clarify this point and provide a prescription for determining the chiral ring relations, from which they deduce properties of the theories at singular points on the Coulomb branch. The authors also discuss the Higgs branch of the theories they study, thus providing a full description of the moduli space. For certain “bad quivers”, they conjecture the existence of new superconformal field theories. The paper ends by discussing mirror symmetry and tests thereof using the $S^3$ partition function. Of particular interest are examples where the duality is between a bad theory and a good one.

I have a very small question: what are the theories that the authors are referring to in Footnote 1? Do they necessarily have non-zero Chern-Simons levels? I think it would be good to be more explicit.

Apart from this, I find the paper to be very clear and comprehensive, and I recommend it for publication after the question asked above is answered.

Requested changes

No changes, apart from the question asked above.

  • validity: top
  • significance: high
  • originality: high
  • clarity: top
  • formatting: perfect
  • grammar: perfect

Author:  Benjamin Assel  on 2018-06-25  [id 279]

(in reply to Report 2 on 2018-06-20)
Category:
answer to question

We thank the referee for reading our paper and for his very positive comments. In footnote 1 we are referring to theories where there is not enough matter to Higgs the gauge group completely, as happens for instance in $U(N)$ SQCD with $N_f<2N$ flavours. These theories fall in the class of ``bad theories", which we study in the paper. We will add a sentence to make the footnote more explicit.

Anonymous on 2018-07-01  [id 284]

(in reply to Benjamin Assel on 2018-06-25 [id 279])
Category:
remark

Thanks to the authors for the explanation. Improving the footnote in the way they suggested sounds good.

I recommend the paper for publication after this is implemented.

---

## Editorial Decision

published